



# Multi-objective calibration of vertical-axis wind turbine controllers: balancing aero-servo-elastic performance and noise

Livia Brandetti*[1,2], Sebastiaan Paul Mulders[2], Roberto Merino-Martinez[3], Simon Watson[1], and Jan-Willem van Wingerden[2]

[1]Flow Physics and Technology, Faculty of Aerospace Engineering, Delft University of Technology, Delft, The Netherlands
[2]Delft Center for Systems and Control, Faculty of Mechanical Engineering, Delft University of Technology, Delft, The Netherlands
[3]Aircraft Noise and Climate Effects, Faculty of Aerospace Engineering, Delft University of Technology, Delft, The Netherlands

**Correspondence:** Livia Brandetti (l.brandetti@tudelft.nl)

**Abstract.** Vertical-axis wind turbines (VAWTs) are considered promising solutions for urban wind energy generation due to their design, low maintenance costs, and reduced noise and visual impact compared to horizontal-axis wind turbines (HAWTs). However, deploying these turbines close to densely populated urban areas often triggers considerable local opposition to wind energy projects. Among the primary concerns raised by communities is the issue of noise emissions. Noise annoyance should be considered in the design and decision-making process to foster the social acceptance of VAWTs in urban environments. For the first time, psychoacoustic annoyance is employed as an indicator to satisfy the noise objective in a controller calibration, as this novel metric provides a more reliable estimate of the human perception of wind turbine noise than conventional sound metrics. At the same time, maximising the operational efficiency of VAWTs in terms of power generation and actuation effort is equally important. This paper addresses the pivotal trade-off between operational performance and noise emissions from a controller calibration perspective. A multi-objective optimisation approach is used to obtain the optimal set of controller calibration solutions balancing the discussed objectives. To maximise the flexibility of controller calibration, the combined wind speed estimator and tip-speed ratio (WSE-TSR) tracking controller is employed as an advanced partial-load control scheme often considered in industrial turbines, and the $K\omega^2$ controller serves as a baseline strategy for comparison. By applying a multi-criteria decision-making method (MCDM), optimal solutions are found that strike a balance between power extraction, actuation effort, and psychoacoustic annoyance. An assessment of these trade-offs, using a frequency-domain framework and mid-fidelity time-domain aero-servo-elastic simulations, yields insights into the meaningful performance metrics of the optimally calibrated WSE-TSR tracking controller. The MCDM results indicate the potential application of this controller in small-scale urban VAWTs to attain power gains of up to 39% on one side and to trade-off a reduction in actuation effort of up to 25% at the cost of only a 2% power decrease and a 6% increase in psychoacoustic annoyance on the other side compared to the baseline. These findings confirm the flexible structure of the optimally calibrated WSE-TSR tracking controller, effectively balancing aero-servo-elastic performance with noise emissions.



# 1 Introduction

The transition from fossil fuels to sustainable energy sources is motivated by the escalating demand for energy and the imperative to curtail greenhouse gas emissions. In this context, wind energy is vital, accounting for $906\,\mathrm{GW}$ of global installed
capacity as of 2022, with an annual growth rate of 9% (Hutchinson and Zhao, 2023). Projections for the next five years anticipate $680\,\mathrm{GW}$ of new installed capacity with an annual growth rate of 13%, considering both onshore and offshore locations. The offshore wind sector has garnered significant attention, primarily due to its abundant wind resources, which can be harnessed by large-scale wind turbines with an average rated output of around $8\,\mathrm{MW}$ connected to the grid (Ramirez, 2023). However, it is worth noting that offshore wind installation is often associated with high costs both in construction and grid
connection, hindering its rapid expansion compared to onshore wind projects (Veers et al., 2019).

Onshore wind sites remain critical for the exploitation of wind energy (Watson et al., 2019). While most of this energy is generated from large-scale turbines (Veers et al., 2019), there is a growing interest in small-scale turbines due to their potential applications in urban environments (Bianchini et al., 2022). Potential integration of small rotors on tall buildings might address local renewable energy demands, complementing the push for sustainable building design (Balduzzi et al., 2012). Moreover, the
importance of small-scale wind turbines extends to future distributed energy networks, especially when effectively combined with energy storage systems (Papi et al., 2021).

In this context, vertical-axis wind turbines (VAWTs) present an attractive opportunity to harness urban wind conditions, characterised by low average wind speeds and high turbulence levels, because of their ability to receive wind from any direction without requiring a yaw mechanism (Mertens et al., 2003), their simple blade design leading to cost-effective mainte-
nance (Howell et al., 2010) and their reduced visual impact (Dayan, 2006; Khan et al., 2017) compared to the horizontal-axis wind turbines (HAWTs) dominating the urban wind energy market.

However, two main challenges remain for the urban deployment of small-scale VAWTs. The first revolves around the need to foster community engagement and social acceptance (Watson et al., 2019; Bianchini et al., 2022). Noise annoyance significantly contributes to the local opposition against urban wind energy projects (Klok et al., 2023). Measures taken to mitigate such
concerns result in reduced power capture efficiency, adversely impacting revenue generation, particularly if the turbines are required to cease operation during nighttime hours (Merino-Martínez et al., 2021). To reduce the impact of such measures on the VAWT performance, an accurate prediction of the wind turbine noise impact on nearby residents is essential. This task is complex due to the influence of various factors, such as wind speed, direction, distance, and background noise (Poulsen et al., 2019). Commonly used time-averaged metrics, such as A-weighted sound pressure level or the day-evening-night level ($L_{\mathrm{den}}$),
may not fully capture the sound properties responsible for noise annoyance (Pieren et al., 2019). Therefore, recent efforts have focused on the auralization of environmental acoustic scenarios. Similar to its visual counterpart, this technique allows for the artificial reproduction of audible situations using numerical data (Vorländer, 2008). A notable contribution to this topic comes from the work of Merino-Martínez et al. (2021), who proposed a novel holistic approach based on synthetic sound auralization and psychoacoustic sound quality metrics to evaluate the annoyance caused by wind turbine noise.



Turning to the second challenge, optimising the controller to ensure optimal performance of small-scale VAWTs in turbulent and fluctuating wind conditions is paramount (Watson et al., 2019; Bianchini et al., 2022). The combined wind speed estimator and tip-speed ratio (WSE-TSR) tracking controller (Bossanyi, 2000) has been successfully applied to maximise the energy capture of VAWTs (Eriksson et al., 2013; Bonaccorso et al., 2011), demonstrating good dynamic performance in tracking the optimal operating point in turbulent wind conditions. This control scheme ensures that the wind turbine operates at the maximum power coefficient associated with a particular tip-speed ratio and pitch angle (Burton et al., 2001). To track the optimal operating point and extract the maximum power, the estimated rotor-effective wind speed (REWS) (Østergaard et al., 2007; Soltani et al., 2013) is used to compute the desired rotor speed reference.

However, the optimal calibration of the WSE-TSR tracking controller is a crucial and nontrivial task due to the controller's nonlinearity and high dependence on a priori model information. The first effort in providing insights into the complex dynamic of the scheme is the derivation of a linear frequency-domain framework in (Brandetti et al., 2022). The work also reveals that the system is ill-conditioned, meaning that the scheme is unable to uniquely provide a wind speed estimate from the product with other internal model parameters. While the frequency-domain framework provides insights for analysing turbine controllers in terms of bandwidth, relating the linear framework to practically meaningful performance metrics (e.g. energy capture and actuation effort) remains an intricate task.

To this end, a recent study by the same authors (Brandetti et al., 2023b) focused on finding the optimal calibration of the WSE-TSR tracking controller. Given the recent interest in and effectiveness of multi-objective optimisation methods based on Pareto fronts (Odgaard et al., 2016; Moustakis et al., 2019; Lara et al., 2023a), the calibration is, therefore, addressed as a multi-objective optimisation problem, with power maximisation and actuation effort minimisation as conflicting objectives. The collection of solutions that are Pareto optimal is then evaluated with a frequency-domain framework to relate performance metrics to controller insights. Results obtained using the NREL 5 MW reference HAWT (Jonkman et al., 2009) under realistic turbulent wind conditions show that when compared to the baseline $K\omega^2$ controller, an optimally calibrated WSE-TSR tracking control strategy does not enhance power capture but does enable the reduction of torque actuation effort with a minor decrease in power production. This finding contradicts the expectations from the existing literature, which claimed energy capture benefits of 1% to 3% when applying a manually calibrated WSE-TSR tracking controller (Holley et al., 1999; Bossanyi, 2000). However, these conclusions were made more than two decades back and based on the application of wind turbines much smaller than the NREL 5 MW turbine.

Hence, validating the above-mentioned hypothesis on a small-scale wind turbine, like an urban VAWT, holds a significant interest. This paper tackles the multi-objective optimisation problem from a control perspective by balancing aero-servo-elastic turbine performance (power capture and actuation effort) with noise (psychoacoustic annoyance). Finding a balance between these objectives will further promote the application of VAWTs in urban environments.

The 1.5 m two-bladed H-Darrieus VAWT (LeBlanc and Simão Ferreira, 2021) is chosen as a case study in the current work. The selection of this specific turbine is motivated by the availability of experimental aerodynamic data and its suitability for rooftop integration (Balduzzi et al., 2012). The psychoacoustic annoyance value needed for the optimal controller calibration is computed by coupling the perception-based approach proposed by Merino-Martínez et al. (2021) and the low-fidelity noise





prediction model developed and validated against high-fidelity simulations by Brandetti et al. (2023a). As experimental acoustic VAWT data are unavailable, the aforementioned model is applied, providing the estimated noise spectra for the small-scale VAWT. These signals are subsequently auralized and assessed with psychoacoustic sound quality metrics to estimate the psychoacoustic noise annoyance.

The optimisation process explores the parameter space of the considered WSE-TSR tracking controller through a guided search procedure. Optimal solutions are identified to form the Pareto front, balancing power maximisation, actuation effort minimisation, and psychoacoustic annoyance minimisation. These optimal results are then evaluated by a linear frequency-domain system and controller analysis framework (Brandetti et al., 2023b) for comparison to the baseline $K\omega^2$ controller. Therefore, the main contributions of this paper are:

- Integrating perception-based psychoacoustic sound quality metrics with a low-fidelity noise prediction model to accurately predict and characterise the acoustic emissions of a small-scale VAWT in terms of psychoacoustic annoyance.

- Presenting an architecture for implementing torque control strategies in small-scale VAWTs with the mid-fidelity software QBlade (Marten, 2020) to conduct realistic aero-servo-elastic simulations of an urban VAWT.

- Formulating and solving a multi-objective optimisation problem for finding the optimal calibration of the WSE-TSR tracking controller as a trade-off between acoustic and aero-servo-elastic performance for an urban VAWT, for the first time taking into account residential concerns in the decision-making process.

The paper is structured as follows: Section 2 derives the model for the wind turbine under study and presents the two considered torque control strategies, namely the WSE-TSR tracking controller and the baseline $K\omega^2$ controller. Section 3 presents the combined noise prediction model and psychoacoustic annoyance model. The architecture for implementing the considered torque control strategies in QBlade and their calibration by means of multi-objective optimisation are provided in Section 4. The optimally calibrated WSE-TSR tracking control scheme is compared to the baseline for its performance in Section 5 using both the frequency-domain framework and the time-domain simulations performed with QBlade. Section 6 offers a summary of the key findings and proposes directions for future work.

**Notations**

This section outlines the notations used in the paper. The notations $\hat{(\cdot)}$ and $\dot{(\cdot)}$ indicate estimated quantities and time derivatives, respectively. The symbol $\bar{(\cdot)}$ represents values corresponding to a specific operating point but also a mean value, whereas $(\cdot)_*$ denotes values indicating the intended optimal or reference parameters.

## 2 Theory and derivations of wind turbine controllers

The $K\omega^2$ controller is an effective and widely used approach for maximising energy capture in partial load operation. However, the control strategy is limited in balancing power and actuation effort objectives for large-scale wind turbines. To address this





issue, the more advanced WSE-TSR tracking scheme offers greater control flexibility. The current study aims to evaluate these findings for small-scale wind turbines, particularly VAWTs, which are promising solutions for urban environments.

This section describes the wind turbine system and its linearised dynamics. Afterwards, the complete and non-linear representations of both the WSE-TSR tracking controller and the baseline $K\omega^2$ used for comparison are derived. This process involves identifying the essential component building blocks for each scheme. Subsequently, a linear frequency-domain framework is formulated to analyse the controllers and closed-loop systems. This framework was derived in (Brandetti et al., 2023b), and the main results are given here; the interested reader is referred to the referenced work for a detailed derivation. The subscripts $(\cdot)_{\text{K}}$ and $(\cdot)_{\text{TSR}}$ differentiate between the transfer functions for the $K\omega^2$ and WSE-TSR control schemes, respectively.

## 2.1 Vertical-axis wind turbine

In this section, the model for the VAWT is presented. Specifically, a two-bladed H-Darrieus turbine is considered, for which experimental aerodynamic data are available, as shown in Figure 1(**a**). To minimise blade deflection, two horizontal struts are used for each blade, located at approximately 25% and 75% of the blade length. The blades have a NACA 0021 profile with a chord length $c_{\text{b}} = 0.075\,\text{m}$, while the struts have a NACA 0018 profile with a chord length $c_{\text{s}} = 0.060\,\text{m}$. The diameter of the VAWT is $D = 1.48\,\text{m}$, with a span $s$ and a height $h$, both equal to $1.5\,\text{m}$. These specifications are summarised in Table 1, and more detailed information about the VAWT design can be found in previous work (LeBlanc and Simão Ferreira, 2021), where the turbine was experimentally investigated.

Figure 1(**b**) shows the turbine Cartesian coordinate system with the origin at the turbine centre. To aid in the interpretation of the results, the blade rotation is divided into two regions: the upwind region, where $0° \leq \theta < 180°$, and the downwind region, where $180° \leq \theta < 360°$. The blade azimuthal position $\theta$ is defined with respect to blade 1, and $\theta = 90°$ and $\theta = 270°$ represent the most upwind and downwind positions, respectively. Blade 2 lags behind blade 1 by $\theta = 180°$.

**Table 1.** PitchVAWT design specifications (LeBlanc and Simão Ferreira, 2021).

| Parameter | Value |
|---|---:|
| Number of blades ($N_b$) | 2 |
| Span ($s$) | $1.5\,\text{m}$ |
| Height ($h$) | $1.5\,\text{m}$ |
| Diameter ($D$) | $1.5\,\text{m}$ |
| Blade chord length ($c_{\text{b}}$) | $7.5 \times 10^{-2}\,\text{m}$ |
| Strut chord length ($c_{\text{s}}$) | $6 \times 10^{-2}\,\text{m}$ |
| Rated power ($P$) | $600\,\text{W}$ |
| Generator efficiency ($\mu$) | 1 |
| Gearbox ratio ($N$) | 1 |
| Rotor inertia ($J$) | $1.5\,\text{kg}\,\text{m}^2$ |





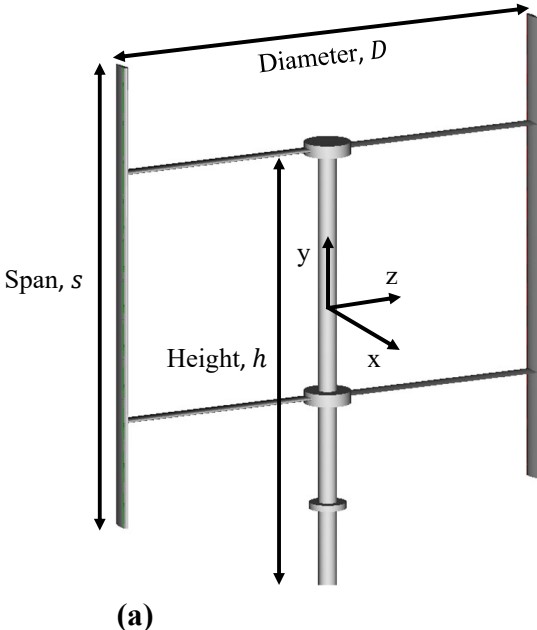

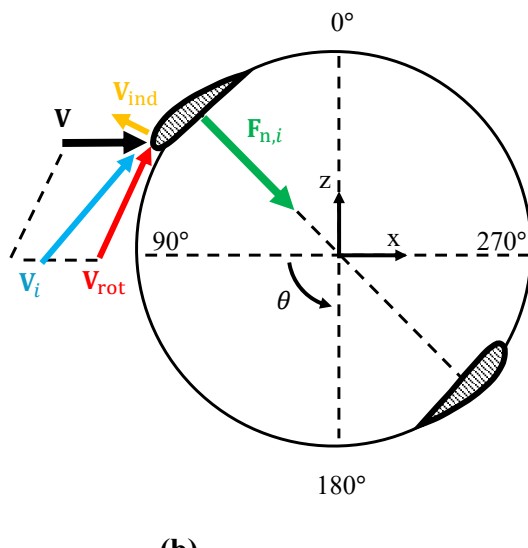

**(a)**          **(b)**

**Figure 1. (a)** Vertical-axis wind turbine (VAWT) geometry and dimensions. The turbine is a two-bladed H-Darrieus VAWT, with diameter $D$, span $s$ and height $h$ of dimensions equal to $1.5\,\mathrm{m}$. **(b)** Coordinate system and definition of the rotor-effective wind speed (REWS) $\mathbf{V}$, the blade-effective wind speed (BEWS) $\mathbf{V}_i$ and the normal load acting on the blade per unit span $F_{\mathrm{n},i}$ vectors adapted from (De Tavernier, 2021). The coordinate system is Cartesian, with the origin at the turbine centre. The blade azimuthal position $\theta$ is defined with respect to blade 1 and is considered positive in the counterclockwise direction. The vector of the BEWS $\mathbf{V}_i$ for the blade $i$ results from the summation of three vector components: the REWS $\mathbf{V}$, the rotational velocity $\mathbf{V}_{\mathrm{rot}}$ of the turbine, and the induced velocity $\mathbf{V}_{\mathrm{ind}}$. The normal force $\mathbf{F}_{\mathrm{n},i}$ per unit span has a positive sign when the vector points inwards.

By examining the 2D blade element depicted in Figure 1**(b)**, the vector of the blade-effective wind speed (BEWS) is defined for each blade as follows:

$$\mathbf{V}_i = \mathbf{V} + \mathbf{V}_{\mathrm{rot}} + \mathbf{V}_{\mathrm{ind}} \,, \tag{1}$$

where $i \in N_{\mathrm{b}} = \{1,2\}$ is the blade index for the VAWT under study, $\mathbf{V}$ denotes the vector for the REWS, $\mathbf{V}_{\mathrm{rot}}$ represents the vector of the tangential velocity of the rotor, resulting from the cross product of the vectors of the rotational speed and

the radius of the turbine, and $\mathbf{V}_{\mathrm{ind}}$ is the vector of the induced velocity, caused by the force field that the turbine generates during the rotation. A detailed derivation of the BEWS can be found in (De Tavernier, 2021) for interested readers. In the following, the italicised notations $V$ and $V_i$ denote the scalar representation for the REWS and BEWS vector quantities $\mathbf{V}$ and $\mathbf{V}_i$, respectively. The wind turbine rotor dynamics are given by

$$J\dot{\omega}_{\mathrm{r}} = T_{\mathrm{r}} - T_{\mathrm{g}} N \,, \tag{2}$$

$$\dot{\theta} = \omega_{\mathrm{r}} \,, \tag{3}$$



where $J$ is the effective low-speed shaft inertia and is derived from the relation $J = J_\mathrm{r} + J_\mathrm{g}N^2$, in which $J_\mathrm{r}$ and $J_\mathrm{g}$ are the inertia of the rotor and generator, respectively, $T_\mathrm{g}$ is the generator torque, and $N := \omega_\mathrm{g}/\omega_\mathrm{r}$ represents the gearbox ratio of the transmission, with $\omega_\mathrm{g}$ and $\omega_\mathrm{r}$ being the generator and the rotor speed, respectively. Assuming a pitch angle $\beta$ constant at an angle of $0°$, a value that maximises the aerodynamic efficiency for the below-rated region, the aerodynamic rotor torque can be formulated as

$$T_\mathrm{r} := \frac{1}{2}\rho A_\mathrm{rot}\frac{V^3}{\omega_\mathrm{r}}C_\mathrm{p}(\lambda,\theta), \tag{4}$$

with $\rho$ and $A_\mathrm{rot}$ being the air density and the rotor area, respectively. In contrast to a HAWT, the power coefficient $C_\mathrm{p}$ for a VAWT is a non-linear mapping in terms of azimuth angle and tip-speed ratio

$$\lambda := \frac{\omega_\mathrm{r}R}{V}, \tag{5}$$

where $R$ represents the rotor radius. This dependency arises from the VAWT operation as the BEWS, and the angle of attack varies with the azimuth rotation angle, resulting in intrinsic three-dimensional aerodynamics (Simão Ferreira et al., 2009). These periodic and non-linear system characteristics are reflected in the dynamics of the VAWT, as illustrated in Equation (3) and Figure 2, where the $C_\mathrm{p}$ curves of the VAWT under study are plotted. The power coefficient mapping exhibits a periodicity of twice-per-revolution (2P) due to the turbine having two blades (Lao et al., 2022).

For a VAWT, the normal load acting on the blade per unit span, shown in Figure 1, is defined as follows:

$$F_{\mathrm{n},i} = \frac{1}{2}\rho c_\mathrm{b}V_i^2 C_{\mathrm{n},i}(\lambda,\theta_i), \tag{6}$$

where $V_i$ represents the magnitude of the BEWS for each blade. The normal load coefficient, denoted as $C_{\mathrm{n},i}$, is a non-linear function that depends on the tip-speed ratio and azimuthal position of blade $i$. It should be noted that $C_{\mathrm{n},i}$ also varies with the

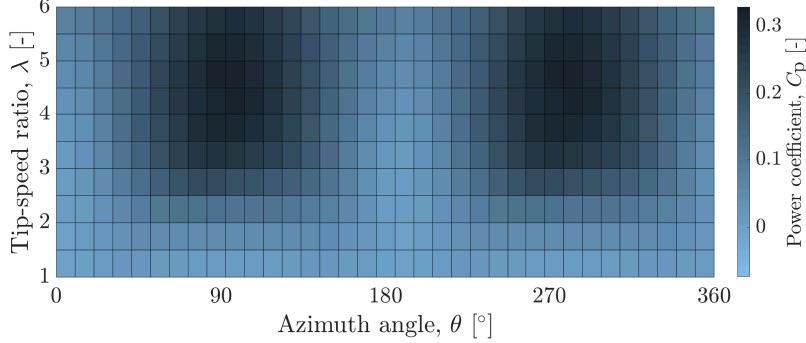

**Figure 2.** Power coefficient $C_\mathrm{p}$, as a function of the tip-speed ratio $\lambda$ and azimuth angle $\theta$, for the two-bladed H-Darrieus VAWT. The maximum values for the $C_\mathrm{p}$ are observed at $\theta = 90°$ and at $\theta = 270°$, as they correspond to the most upwind locations for blade 1 and blade 2, respectively. Due to the presence of these two blades, the twice-per-revolution (2P) periodicity of $C_\mathrm{p}$ is evident, especially at high values of $\lambda$.



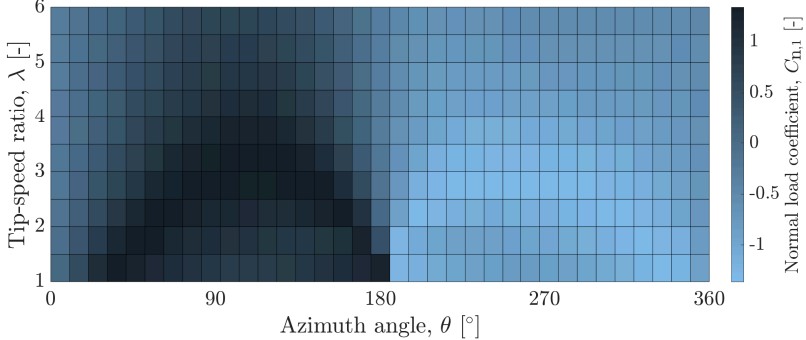

**Figure 3.** Normal load coefficient mapping $C_{\mathrm{n},1}$, as a function of the tip-speed ratio $\lambda$ and azimuth angle $\theta$, for blade 1 of the H-Darrieus VAWT. It is evident that the normal blade force varies over the rotation, being positive upwind ($0° \leq \theta < 180°$) and negative downwind ($180° \leq \theta < 360°$), with its maximum value at an azimuth angle $\theta = 90°$. This dynamics demonstrates the presence of a once-per-revolution periodicity (1P) on the $C_{\mathrm{n},1}$.

blade pitch angle $\beta_i$. However, $\beta_i$ is considered constant throughout this study, maintaining a value of $0°$. Figure 3 illustrates
the $C_{\mathrm{n},1}$ curve for the VAWT. Notably, a maximum normal load occurs at approximately $\theta = 90°$, corresponding to blade 1
being upwind. Blade 2 exhibits similar behaviour, although with a $180°$ shift. The variations in load dynamics throughout the
rotation demonstrate the presence of a once-per-revolution periodicity (1P) in $C_{\mathrm{n},i}$.

The wind turbine can be linearised around a specific operating point with the definition of the rotor and blade dynamics at
hand. Firstly, the non-linear formulation of the aerodynamic rotor torque from Equation (4) is combined with Equation (2).
The resultant expression is then linearised concerning the rotor speed state, wind speed disturbance input, and generator torque
control input. The outcome is represented by

$$\dot{\omega}_{\mathrm{r}} = G(V)\,\omega_{\mathrm{r}} + H(V)\,V + E\,T_{\mathrm{g}}.\tag{7}$$

The original variables express the values perturbed around their operating points to ensure conciseness, while $G(V), H(V)$
and $E$ represent partial derivatives defined as

$$180 \quad G(V) = \frac{1}{J}\left.\frac{\partial T_{\mathrm{r}}}{\partial \omega_{\mathrm{r}}}\right|_{(\bar{\omega}_{\mathrm{r}},\bar{V})} = \frac{1}{2J}\rho A_{\mathrm{rot}}\left(-\frac{V^3}{\omega_{\mathrm{r}}^2}C_{\mathrm{p}}(\omega_{\mathrm{r}},V) + \frac{V^2 R}{\omega_{\mathrm{r}}}\frac{\partial C_{\mathrm{p}}(\omega_{\mathrm{r}},V)}{\partial \lambda}\right)\bigg|_{(\bar{\omega}_{\mathrm{r}},\bar{V})},\tag{8}$$

$$H(V) = \frac{1}{J}\left.\frac{\partial T_{\mathrm{r}}}{\partial V}\right|_{(\bar{\omega}_{\mathrm{r}},\bar{V})} = \frac{1}{2J}\rho A_{\mathrm{rot}}\left(\frac{3V^2}{\omega_{\mathrm{r}}}C_{\mathrm{p}}(\omega_{\mathrm{r}},V) - VR\frac{\partial C_{\mathrm{p}}(\omega_{\mathrm{r}},V)}{\partial \lambda}\right)\bigg|_{(\bar{\omega}_{\mathrm{r}},\bar{V})}, \quad E = -\frac{N}{J}.\tag{9}$$

The variable $V$ is introduced to conveniently define estimator-based expressions for $G$ and $H$ in a subsequent section, but it is
excluded in the following terms.



The time domain form of Equation (7) is then Laplace transformed to give the transfer functions from wind speed and generator torque inputs to rotor speed as output

$$\Omega_{\mathrm{r}}(s) = \frac{H}{s - G}\mathcal{V}(s) + \frac{E}{s - G}\mathcal{T}_{\mathrm{g}}(s),\tag{10}$$

where $s$ represents the Laplace operator. The variables $\Omega_{\mathrm{r}}, \mathcal{V}$, and $\mathcal{T}_{\mathrm{g}}$ indicate the frequency-domain representation of the rotational speed, wind speed, and generator torque, respectively.

## 2.2 Torque control strategies

The two torque control strategies applied to the VAWT are formalised in the following based on the defined wind turbine dynamics. First, the complete and non-linear formulation of the $K\omega^2$ controller is obtained, followed by the derivation of the WSE-TSR tracking controller. An overview of the control frameworks is provided in Figure 4 to facilitate the comparison between the two schemes. As can be observed, both controllers aim to maximise the power production of the urban VAWT by using the reference tip-speed ratio $\lambda_*$ and the measured rotor speed $\omega_{\mathrm{r}}$ as inputs. A more detailed description of each block diagram is given in the sub-sections below.

### 2.2.1 Baseline $K\omega^2$ controller

The $K\omega^2$ controller is widely used for the operation of a small-scale VAWT. As Haque et al. (2008) demonstrated, this controller effectively maximises turbine power production by measuring the rotor speed and determining the reference torque. However, no comparison with a more advanced controller is provided. This study employs the $K\omega^2$ control law as a baseline, deriving the equations characterising its performance. The block diagram of the controller, shown in Figure 4 **(a)**, includes the wind turbine and the controller. It becomes evident that the controller operates as a static non-linear function, generating the control signal for the generator torque using the rotor speed. The control signal is given by

$$T_{\mathrm{g,K}} = K\frac{\omega_r^2}{N},\tag{11}$$

where the torque gain $K$ (Bossanyi, 2000) is determined at the low-speed shaft side of the drivetrain as

$$K = \frac{\rho A_{\mathrm{rot}} R^3 C_{\mathrm{p},*}(\lambda_*)}{2\lambda_*^3}.\tag{12}$$

The optimal power coefficient for maximum energy extraction and the associated optimal tip-speed ratio are $C_{\mathrm{p},*}$ and $\lambda_*$, respectively.

### 2.2.2 WSE-TSR tracking controller

The WSE-TSR tracking control scheme illustrated in Figure 4 **(b)** comprises an estimator and a tip-speed ratio tracking controller and has been shown to optimise the turbine performance of VAWTs in turbulent wind conditions (Eriksson et al., 2008; Bonaccorso et al., 2011). The estimator employs the measured output of the real system, control signal, and a non-linear wind



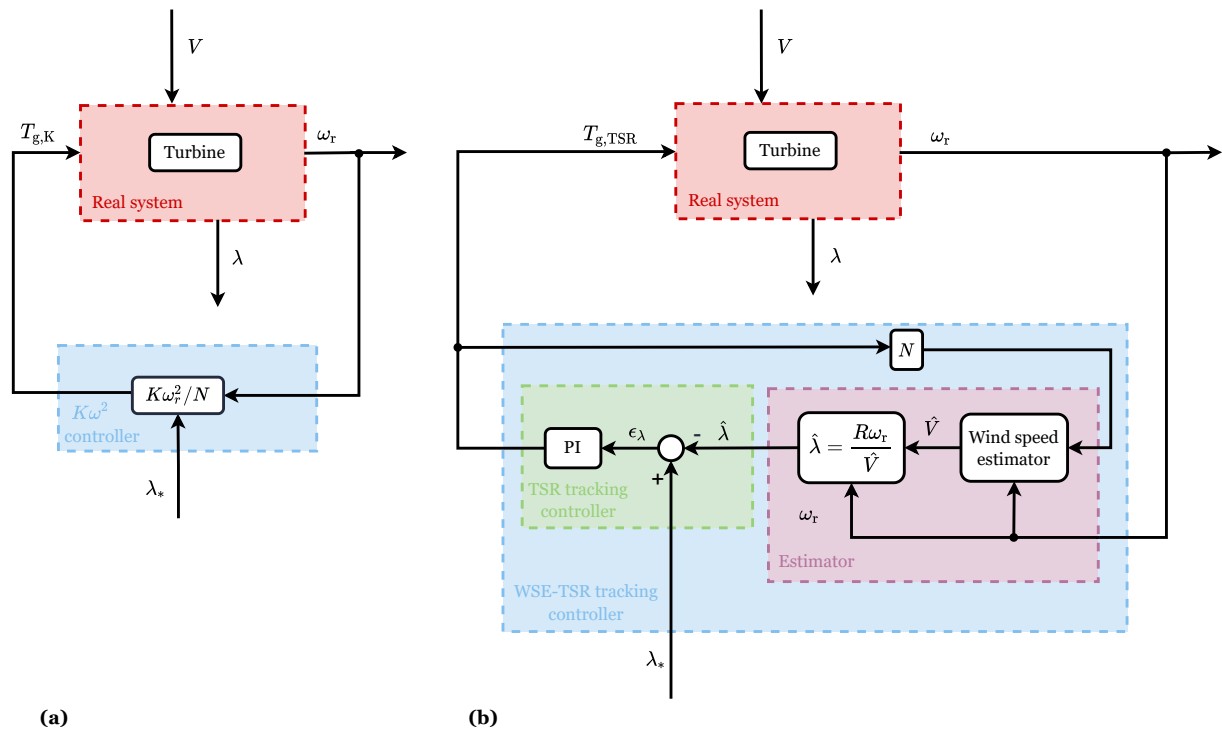

**(a)**                                    **(b)**

**Figure 4.** Block diagram for **(a)** the $K\omega^2$ and **(b)** the WSE-TSR tracking control frameworks. In both schemes, the wind turbine system, highlighted in the **red** box, has two inputs (the wind speed, $V$, and the generator torque, $T_{g,K}$ and $T_{g,TSR}$ respectively if the $K\omega^2$ or the WSE-TSR tracking controller is applied), and two outputs (the tip-speed ratio, $\lambda$, and the rotational speed, $\omega_r$). In **(a)** the $K\omega^2$ block diagram, the controller (**cyan** box) uses the measured $\omega_r$ and the optimal tip-speed ratio, $\lambda_*$, as inputs to compute $T_{g,K}$. On the other hand, for **(b)** the WSE-TSR tracking controller, the **cyan** box encompasses the estimator (**purple** box) and the TSR tracker controller (**green** box). The estimator block applies the measured $T_{g,TSR}$ and $\omega_r$ to estimate the rotor-effective wind speed $\hat{V}$ and to calculate an estimate of the tip-speed ratio, $\hat{\lambda}$. The controller operates on the difference between $\hat{\lambda}$ and the optimal tip-speed ratio, $\lambda_*$, to determine the torque control signal $T_{g,TSR}$.

turbine model to compute an estimate of the REWS using the Immersion and Invariance (I&I) estimator (Ortega et al., 2013) with an augmented integral correction term (Liu et al., 2022). Assuming the measurement of the generator torque control input and the turbine's rotational speed and considering the REWS as an unknown positive disturbance input to the plant, the
formulation of the estimator is as follows

$$
\begin{cases}
J\dot{\hat{\omega}}_r = \hat{T}_r - T_{g,TSR}N \\
\epsilon_{\omega_r} = \omega_r - \hat{\omega}_r \\
\hat{V} = K_{p,w}\epsilon_{\omega_r} + K_{i,w}\int_0^t \epsilon_{\omega_r}(\tau)\mathrm{d}\tau
\end{cases}
,
\tag{13}
$$





where $\hat{V}$ represents the estimated REWS, $K_{\mathrm{p,w}}$ and $K_{\mathrm{i,w}}$ are the proportional and integral estimator gains, respectively, $t$ denotes the current timestep, and $\tau$ is the integration variable. The estimated aerodynamic torque is defined as

$$\hat{T}_{\mathrm{r}} = \frac{1}{2}\rho A_{\mathrm{rot}}\frac{\hat{V}^3}{\omega_{\mathrm{r}}}\hat{C}_{\mathrm{p}}(\hat{\lambda}),\tag{14}$$

with $\hat{C}_{\mathrm{p}}$ being the estimated power coefficient, a non-linear mapping of the estimated tip-speed ratio $\hat{\lambda} = \omega_{\mathrm{r}}R/\hat{V}$. Also, in this case, the pitch angle $\beta$ is constant and equals $0°$.

Then, the proportional and integral (PI) controller in the WSE-TSR tracking scheme operates on the difference between the estimated and reference tip-speed ratio $\lambda_*$. The resulting error is utilised to determine $T_{\mathrm{g,TSR}}$, being the generator torque demand, forcing the turbine to track the reference as

$$225 \quad T_{\mathrm{g,TSR}} = K_{\mathrm{p,c}}\epsilon_\lambda + K_{\mathrm{i,c}}\int_0^t \epsilon_\lambda(\tau)\mathrm{d}\tau,\tag{15}$$

in which $\epsilon_\lambda$ is the tip-speed ratio error, $K_{\mathrm{p,c}}$ is the proportional controller gain and $K_{\mathrm{i,c}}$ is the integral controller gain.

## 2.3 Analysis framework

The universal analysis framework proposed in (Brandetti et al., 2023b) is used to evaluate the characteristics of the described control strategies and closed-loop systems. Only the main results are given in this section; the reader is referred to the referenced 230 work for a more extensive derivation and explanation of the framework. The framework is depicted in Figure 5, where the

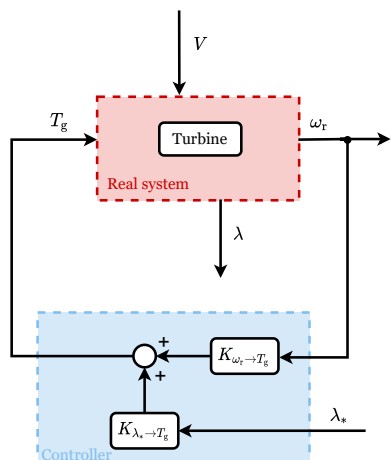

**Figure 5.** Block diagram illustrating the universal framework employed for the controller analysis (Brandetti et al., 2023b). The wind turbine system is represented with the **red** box having the generator torque, $T_{\mathrm{g}}$, and the wind speed, $V$, as inputs, and the rotational speed, $\omega_{\mathrm{r}}$, and the tip-speed ratio, $\lambda$, as outputs. The feedback term, $K_{\omega_{\mathrm{r}}\to T_{\mathrm{g}}}$, and the reference shaping term, $K_{\lambda_*\to T_{\mathrm{g}}}$, used in the analysis framework, are included in the **cyan** box symbolising the controller with two inputs ($\omega_{\mathrm{r}}$ and the tip-speed ratio set point, $\lambda_*$), one output ($T_{\mathrm{g}}$).





controllers are represented as one block with the rotor speed and the reference tip-speed ratio as inputs and the generator torque control signal as output. Each control scheme is formalised in the linear and frequency-domain formulation as

$$\mathcal{T}_\mathrm{g}(s) = K_{\Omega_\mathrm{r} \to \mathcal{T}_\mathrm{g}}(s)\Omega_\mathrm{r}(s) + K_{\Lambda_* \to \mathcal{T}_\mathrm{g}}(s)\Lambda_*(s),\tag{16}$$

in which $K_{\Omega_\mathrm{r} \to \mathcal{T}_\mathrm{g}}$ and $K_{\Lambda_* \to \mathcal{T}_\mathrm{g}}$ represent the feedback and the reference shaping terms, respectively, and $\Lambda_*$ indicates the reference tip-speed ratio signal in the frequency domain.

Combining Equation (16) with Equation (10) and after manipulation, it is possible to derive the closed-loop transfer functions. As the scheme intends to regulate the tip-speed ratio to its reference value, the closed-loop transfer functions are expressed as a function of the turbine's actual tip-speed ratio $\lambda$. It follows that the two transfer functions representing the closed-loop system reference tracking and disturbance attenuation capabilities are defined as

$$\Lambda(s) = \underbrace{\frac{REK_{\Lambda_* \to \mathcal{T}_\mathrm{g}}(s)}{\bar{V}\left(s - G - EK_{\Omega_\mathrm{r} \to \mathcal{T}_\mathrm{g}}(s)\right)}}_{T_{\Lambda_* \to \Lambda}(s)}\Lambda_*(s) + \underbrace{\frac{R\left(H - (\bar{\omega}_\mathrm{r}/\bar{V})\left(s - G - EK_{\Omega_\mathrm{r} \to \mathcal{T}_\mathrm{g}}(s)\right)\right)}{\bar{V}\left(s - G - EK_{\Omega_\mathrm{r} \to \mathcal{T}_\mathrm{g}}(s)\right)}}_{T_{\mathcal{V} \to \Lambda}(s)}\mathcal{V}(s).\tag{17}$$

Specifically, the term $T_{\Lambda_* \to \Lambda}(s)$ is the complementary sensitivity function, indicating the controller performance in tracking the commanded reference (i.e. $\lambda = \lambda_*$); the sensitivity function $T_{\mathcal{V} \to \Lambda}(s)$ represents the controller performance in rejecting wind speed disturbances (Brandetti et al., 2023b). The $\bar{(\cdot)}$ represents values corresponding to a specific operating point in the analysis framework.

### 2.3.1 Baseline $K\omega^2$ controller

To determine the $K\omega^2$ controller dynamics, Equation (11) can be linearised and combined with Equation (10), representing the linearised wind turbine dynamics. Follows the feedback and the reference shaping terms defined according to the universal controller framework (Brandetti et al., 2023b) as

$$K_{(\Omega_\mathrm{r} \to \mathcal{T}_\mathrm{g}),\mathrm{K}} = \left.\frac{\partial T_{\mathrm{g},\mathrm{K}}}{\partial \omega_\mathrm{r}}\right|_{(\bar{\omega}_\mathrm{r},\lambda_*)} = \frac{2K\bar{\omega}_\mathrm{r}}{N} = \frac{\rho R^3 A_\mathrm{rot} C_{\mathrm{p},*}(\lambda_*)}{N\lambda_*^3}\bar{\omega}_\mathrm{r},\tag{18}$$

$$K_{(\Lambda_* \to \mathcal{T}_\mathrm{g}),\mathrm{K}} = \left.\frac{\partial T_{\mathrm{g},\mathrm{K}}}{\partial \lambda_*}\right|_{(\bar{\omega}_\mathrm{r},\lambda_*)} = \frac{\rho R^3 A_\mathrm{rot}}{2N}\left(-\frac{3}{\lambda_*^4}C_{\mathrm{p},*}(\lambda_*) + \frac{1}{\lambda_*^3}\frac{\partial C_{\mathrm{p},*}(\lambda_*)}{\partial \lambda_*}\right)\bar{\omega}_\mathrm{r}^2.\tag{19}$$

It can be observed from Equations (18) and (19) that the transfer functions of the controller are frequency-independent gains for the baseline controller.

### 2.3.2 WSE-TSR tracking controller

The WSE-TSR tracking controller dynamics are obtained through the initial linear derivation of the individual estimator and controller in the frequency domain. Subsequently, coupling between the estimator and the controller is performed to achieve the dynamics of the overall scheme. The interested reader is referred to Brandetti et al. (2023b) for the complete derivation.





For conciseness, only the controller transfer functions are reported here as

$$K_{(\Omega_r \to \mathcal{T}_g),\mathrm{TSR}}(s) = \frac{\mathcal{T}_{g,\mathrm{TSR}_{\Omega_r}}(s)}{\Omega_r(s)} = \frac{R\left(K_{\mathrm{p,c}}\,s + K_{\mathrm{i,c}}\right)\left(\left(\bar{\omega}_r K_{\mathrm{p,w}} - \bar{V}\right)s^2 + F_4\,s - \left(\bar{V}\hat{H} + \bar{\omega}_r \hat{G}\right)K_{\mathrm{i,w}}\right)}{\left(\bar{V}^2\,s^3 + F_1\,s^2 + F_2\,s + F_3\right)}, \tag{20}$$

and

$$K_{(\Lambda_* \to \mathcal{T}_g),\mathrm{TSR}}(s) = \frac{\mathcal{T}_{g,\mathrm{TSR}_{\Lambda_*}}(s)}{\Lambda_*(s)} = \frac{\bar{V}^2\left(K_{\mathrm{p,c}}\,s + K_{\mathrm{i,c}}\right)\left(s^2 + \hat{H}\,K_{\mathrm{p,w}}\,s + \hat{H}\,K_{\mathrm{i,w}}\right)}{\left(\bar{V}^2\,s^3 + F_1\,s^2 + F_2\,s + F_3\right)}, \tag{21}$$

characterising, on the one hand, the transfer function from the rotational speed to the generator torque output and, on the other hand, the transfer function from the tip-speed ratio reference to the generator torque output. To simplify Equations (20) and (21), the unspecified variables in the preceding formulations are denoted as

$$F_1 = \bar{V}^2 \hat{H} K_{\mathrm{p,w}} + R\bar{\omega}_r E K_{\mathrm{p,c}} K_{\mathrm{p,w}},$$

$$F_2 = \bar{V}^2 \hat{H} K_{\mathrm{i,w}} + R\bar{\omega}_r E K_{\mathrm{p,c}} K_{\mathrm{i,w}} + R\bar{\omega}_r E K_{\mathrm{i,c}} K_{\mathrm{p,w}},$$

$$F_3 = R\bar{\omega}_r E K_{\mathrm{i,c}} K_{\mathrm{i,w}},$$

$$F_4 = \bar{\omega}_r K_{\mathrm{i,w}} - \left(\bar{V}\hat{H} + \bar{\omega}_r \hat{G}\right)K_{\mathrm{p,w}}.$$

As the considered WSE-TSR tracking control scheme incorporates turbine model information that accurately reflects the characteristics of the wind turbine system without explicitly addressing inherent uncertainties present in real-world turbine dynamics, the variables $\hat{G} := G(\hat{V})$ and $\hat{H} := H(\hat{V})$ represent the estimated partial derivatives as formulated in Equations (8) and (9).

## 3 Methodology to assess the noise levels and psychoacoustic annoyance on a VAWT

This section outlines the methodology for estimating the noise generated by the VAWT under investigation and assessing the subsequent expected psychoacoustic annoyance. Figure 6 shows the subsequent steps required to determine the psychoacoustic annoyance metric. First, the acoustic emissions of the VAWT are modelled using the noise prediction method, which was introduced and validated against high-fidelity simulations in Brandetti et al. (2023a). The estimated wind turbine noise spectra over time are then auralized to make the signal audible and then evaluated with a perception-based approach (Merino-Martínez et al., 2021) to determine the expected psychoacoustic annoyance.

### 3.1 Noise prediction model

This section provides an overview of the model used to estimate the aeroacoustics performance of the VAWT. Interested readers are referred to Brandetti et al. (2023a) for more comprehensive details. The model evaluates three distinct noise generation mechanisms that are considered dominant for the two-bladed $1.5\,\mathrm{m}$ H-Darrieus VAWT:

1. Laminar Boundary Layer-Vortex Shedding (LBL-VS) noise;





2.  Turbulent Boundary Layer-Trailing Edge (TBL-TE) noise;

3.  Turbulence-Interaction (T-I) noise.

Among these sources, LBL-VS and TBL-TE are self-generated by the airfoil interacting with a steady flow (Brooks et al., 1989), whereas the T-I noise occurs from the interaction between the blade leading edge and inflow turbulence (Rogers et al., 2006; Kim et al., 2016). Note that the noise prediction model does not consider blade-blade interaction, as the blades are treated as isolated entities, and it assumes steady, free-stream conditions with quasi-steady time dependence.

The estimation of these noise sources involves several steps. Firstly, each blade is discretised in a three-dimensional space, dividing it into sequential strips. These strips possess identical airfoil chords and finite spans. Then, the computational domain is discretised in time, enabling the blade to progress along its rotational trajectory for a complete revolution (Botha et al., 2017). Consequently, for each blade element and azimuthal position, the airfoil-self noise and T-I noise are estimated using the methodologies presented by Brooks et al. (1989) and Buck et al. (2016), respectively. The relevant equations to imple-

ment these models are explained in the following and rely on flow input parameters, including the angle of attack $\alpha$ and the BEWS. In the work conducted by Brandetti et al. (2023a), these parameters were estimated with the two-dimensional Actuator Cylinder model (Madsen, 1982). This study aims to enhance the accuracy of acoustic predictions by solving the flow over the blade using three-dimensional lifting line-free vortex wake simulations performed in the aero-servo-elastic software QBlade (Marten, 2020). After determining the sound pressure levels from these semi-empirical models, a Doppler correction

factor is computed for the considered noise sources to account for the relative motion between the blade and the stationary observer (Ruijgrok, 1993). The total noise emissions along the blades and throughout a single rotation are finally calculated employing the approach Brooks and Burley (2004) developed.

The resulting sound pressure levels for the three noise sources are then used as inputs to perform the sound auralization to make the signals audible and assessable with the perception-based approach to determine the corresponding psychoacoustic

annoyance.

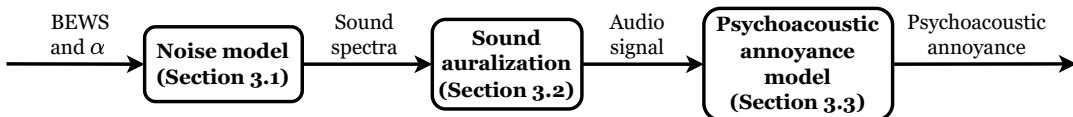

**Figure 6.** Block diagram illustrating the integrated low-fidelity noise model and the psychoacoustic annoyance model based on perception-based psychoacoustic sound quality metrics. The first step is loading the blade-effective wind speed (BEWS) and the angle of attack ($\alpha$), retrieved from the aero-servo-elastic simulations of the VAWT in QBlade (Marten, 2020), into the noise model. Then, the estimated sound spectra are auralized, generating realistic audio signals. These audio files are then evaluated using psychoacoustic sound quality metrics to estimate the psychoacoustic annoyance.





### 3.1.1 Airfoil-self noise model

Only two of five airfoil-self noise mechanisms are considered relevant for a VAWT in an urban environment: the LBL-VS noise and the TBL-TE noise. These noise sources are modelled with the Brooks Pope and Marcolini (BPM) approach (Brooks et al., 1989) and distinguished according to the flow conditions.

LBL-VS noise is dominant at low Reynolds numbers (i.e. $Re \leq 5 \times 10^5$) when Tollmien-Schlichting (T-S) waves develop, leading to the generation of vortex shedding and, consequently, tonal noise through a feedback loop (Brooks et al., 1989). The Sound Pressure Level in $1/3^{\text{rd}}$ octave bands for this noise generation mechanism ($\text{SPL}_{\text{LBL}-\text{VS}}$) is calculated as:

$$\text{SPL}_{\text{LBL}-\text{VS}} = 10 \log_{10}\left(\frac{\delta_{\text{p}} M^5 d \bar{D}_{\text{h}}}{r_{\text{e}}^2}\right) + Q_1\left(\frac{St'}{St'_{\text{peak}}}\right) + Q_2\left[\frac{Re}{(Re)_0}\right] + Q_3(\alpha), \tag{22}$$

where $\delta_{\text{p}}$ is the boundary layer thickness at the pressure side, $d$ is the span-wise size of the blade element, $\bar{D}_{\text{h}}$ is the directivity

function for the high-frequency limit, $r_{\text{e}}$ is the absolute distance to the receiver, and $(Re)_0$ is the chord-based Reynolds number at $\alpha = 0°$. A detailed description of the Strouhal contributions, $St'$ and $St'_{\text{peak}}$, and the empirical functions, $Q_1$, $Q_2$ and $Q_3$ can be found in Brooks et al. (1989).

The boundary layer developing over the airfoil for higher Reynolds numbers (i.e. $Re \geq 5 \times 10^5$) becomes turbulent. These turbulent pressure fluctuations are scattered as TBL-TE noise when convecting over the sharp trailing edge. For estimating

the SPL of this noise source, three contributions are taken into account in the BPM model: one from the attached TBL on the pressure side ($\text{SPL}_{\text{p}}$), one from the attached TBL on the suction side ($\text{SPL}_{\text{s}}$), and a third component accounting for separation-stall at high angles of attack ($\text{SPL}_{\alpha}$). The SPL in $1/3^{\text{rd}}$ octave bands for the TBL-TE noise ($\text{SPL}_{\text{TBL}-\text{TE}}$) is defined as:

$$\text{SPL}_{\text{TBL}-\text{TE}} = 10 \log_{10}\left(10^{\left(\frac{\text{SPL}_{\text{p}}}{10}\right)} + 10^{\left(\frac{\text{SPL}_{\text{s}}}{10}\right)} + 10^{\left(\frac{\text{SPL}_{\alpha}}{10}\right)}\right). \tag{23}$$

$$\text{SPL}_{\text{p}} = \begin{cases} 10 \log_{10}\left(\frac{\delta_{\text{p}}^* M^5 d \bar{D}_{\text{h}}}{r_{\text{e}}^2}\right) + L\left(\frac{St_{\text{p}}}{St_1}\right) + (Z_1 - 3) + \Delta Z_1, & \text{for} \quad \alpha \leq 12.5°, \\ -\infty, & \text{for} \quad \alpha \geq 12.5°, \end{cases} \tag{24}$$

$$\text{SPL}_{\text{s}} = \begin{cases} 10 \log_{10}\left(\frac{\delta_{\text{s}}^* M^5 d \bar{D}_{\text{h}}}{r_{\text{e}}^2}\right) + L\left(\frac{St_s}{St_1}\right) + (Z_1 - 3), & \text{for} \quad \alpha \leq 12.5°, \\ -\infty, & \text{for} \quad \alpha \geq 12.5°, \end{cases} \tag{25}$$

$$\text{SPL}_{\alpha} = \begin{cases} 10 \log_{10}\left(\frac{\delta_s^* M^5 d \bar{D}_{\text{h}}}{r_{\text{e}}^2}\right) + U\left(\frac{St_s}{St_2}\right) + Z_2, & \text{for} \quad \alpha \leq 12.5°, \\ 10 \log_{10}\left(\frac{\delta_s^* M^5 d \bar{D}_{\text{l}}}{r_{\text{e}}^2}\right) + L'\left(\frac{St_s}{St_2}\right) + Z_2, & \text{for} \quad \alpha \geq 12.5°, \end{cases} \tag{26}$$

with $\delta_{\text{p}}^*$ and $\delta_{\text{s}}^*$ being the boundary layer displacement thickness at the pressure side and at the suction side, respectively, $\bar{D}_{\text{l}}$ being the directivity function for the low-frequency limit and $M$ being the free-stream Mach number. For details on the





Strouhal contributions, $St_\mathrm{p}$, $St_\mathrm{s}$, $St_1$ and $St_2$, the empirical functions, $L, L'$ and $U$, and the amplitude correction factors, $Z_1$, $Z_2$ and $\Delta Z_1$, the reader can refer to Brooks et al. (1989).

### 3.1.2 Turbulence-Interaction noise model

Aerodynamic noise caused by the interaction between the incoming turbulent inflow and the leading edge of the blades is commonly referred to as T-I noise (Rogers et al., 2006). In the noise prediction model, this source is modelled with the approach of Buck et al. (2016). The SPL of the T-I noise ($SPL_\mathrm{T-I}$) is computed in 1/3$^{\mathrm{rd}}$ octave bands as the sum of the high-frequency and low-frequency components of the noise

$$SPL_\mathrm{T-I} = SPL_\mathrm{T-I}^{H} + 10\log_{10}\left(\frac{\mathrm{LFC}}{1+\mathrm{LFC}}\right). \tag{27}$$

In the above expression, LFC is the blending function introduced by Lowson and Ollerhead (1969) and Moriarty and Migliore (2003), and $SPL_\mathrm{T-I}^{H}$ is the high-frequency component defined as

$$SPL_\mathrm{T-I}^{H} = 10\log_{10}\left[\frac{\rho^2 c_0^2 d}{2r_e^2}M^3\varepsilon^{(2/3)}k^{-(5/3)}\bar{D}_\mathrm{LE}\right]+77.6\,, \tag{28}$$

where $c_0$ is the sound speed, $k$ is the wave-number ($k = (2\pi f)/V_\mathrm{i}$), and $\bar{D}_\mathrm{LE}$ is the directivity function accounting for the motion between the leading edge and the stationary observer.

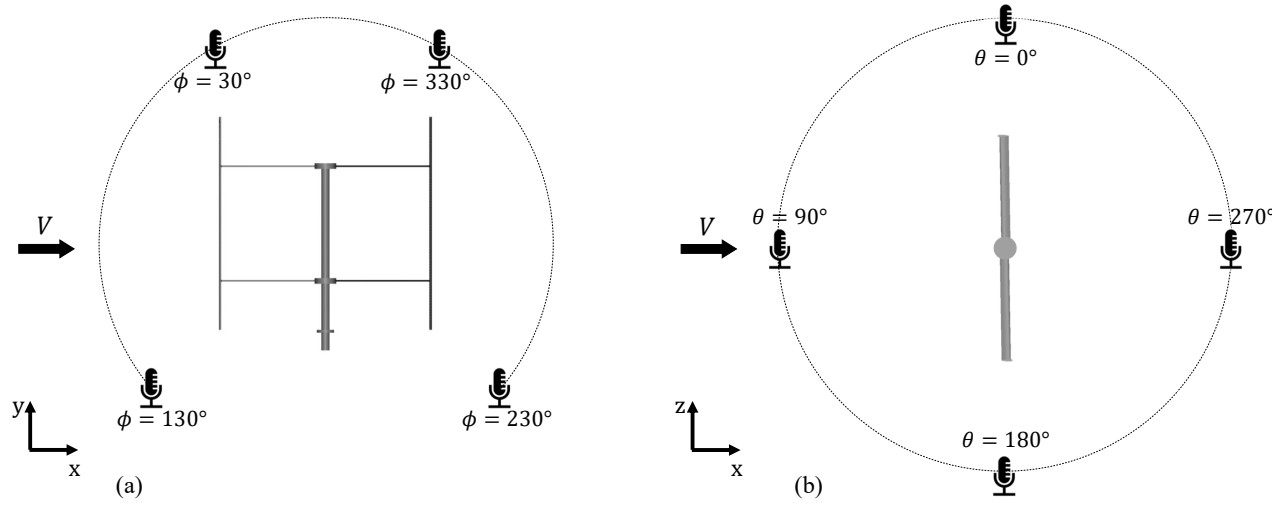

**Figure 7.** Two circular arrays of virtual microphones are positioned at a distance of $2.6D$ from the centre of the VAWT. Array (a) comprises 4 microphones located in the x-y plane, while array (b) consists of 4 microphones positioned in the x-z plane. These locations are considered relevant for characterising the psychoacoustic annoyance of the sound source.




### 3.1.3 Observer location

In the proposed low-fidelity noise model, 8 virtual microphones are considered. As shown in Figure 7, two circular arrays in the x-y and x-z planes, respectively, consisting of 4 virtual microphones, each are positioned at $2.6D$ from the centre of the VAWT. This setup is chosen to cover the three-dimensional sound field of the turbine. The noise model is able to estimate the sound spectra at each of these observer locations. For every case study, a single observer location is chosen in the next section describing the sound auralization procedure.

### 3.2 Sound auralization

The propagated sound spectra estimated in Section 3.1 are auralized to obtain the audio time signal that a virtual observer at a specific location would perceive.

Auralization is a technique that enables artificially making an acoustic situation audible from numerical (simulated, measured or synthesised) data (Vorländer, 2008). It can be considered the acoustic counterpart to visualisation.

To achieve more realistic auralized audio waves, each propagated 1/3$^{\text{rd}}$-octave band spectrum was interpolated to obtain an equivalent narrow-band spectrum with the same sound pressure level per band. The narrow-band spectra are then converted to the time domain using an inverse short-time Fourier transform, following the guidelines explained in (Vorländer, 2008; Merino-Martínez et al., 2021). Each time block was windowed using a Hanning weighting function with 50% data overlap. The resulting audio files were then fed into the psychoacoustic annoyance model (see section Section 3.3) to estimate the psychoacoustic annoyance of each sound which is described in the next section.

### 3.3 Psychoacoustic annoyance model

This section introduces the psychoacoustic annoyance model employed for estimating the perceived annoyance due to the noise emitted by a VAWT in an urban environment. The audio files determined with the auralization of the sound spectra in Section 3.2 are assessed using a combination of Sound Quality Metrics (SQMs) (Di et al., 2016) to estimate the psychoacoustic annoyance.

Unlike the SPL metric, which quantifies the purely physical magnitude of sound, Sound Quality Metrics (SQMs) describe the subjective perception of sound by human hearing. Therefore, these metrics have been shown to better capture the auditory behaviour of the human ear compared to the conventional sound metrics typically employed in wind turbine noise assessments (Merino-Martínez et al., 2021). The psychoacoustic annoyance model considers the five most common SQMs (Greco et al., 2023b):

- Loudness ($\mathcal{N}$): Subjective perception of sound magnitude corresponding to the overall sound intensity (International Organization for Standardization, 2017).

- Tonality ($\mathcal{K}$): Measurement of the perceived strength of unmasked tonal energy within a complex sound (Aures, 1985).

- Sharpness ($\mathcal{S}$): Representation of the high-frequency sound content (von Bismarck, 1974).





- Roughness ($\mathcal{R}$): Hearing sensation caused by sounds with modulation frequencies between $15\,\mathrm{Hz}$ and $300\,\mathrm{Hz}$ (Daniel and Webber, 1997).

- Fluctuation strength ($\mathcal{F}_\mathrm{s}$): Assessment of slow fluctuations in loudness with modulation frequencies up to $20\,\mathrm{Hz}$, with maximum sensitivity for modulation frequencies around $4\,\mathrm{Hz}$ (Osses Vecchi et al., 2016).

All SQMs were computed using the open-source MATLAB toolbox SQAT (Sound Quality Analysis Toolbox) (Greco et al., 2023a) and the 5% percentiles were considered, representing the threshold value of each SQMs that is exceeded for 5% of the total signal time. Following the formulation of Di et al. (2016), these SQMs can be combined to compute the psychoacoustic annoyance (PA) as:

$$\mathrm{PA} = \mathcal{N}\left(1 + \sqrt{v_\mathcal{S}(\mathcal{N},\mathcal{S})^2 + v_{\mathcal{FR}}(\mathcal{N},\mathcal{F}_\mathrm{s},\mathcal{R})^2 + v_\mathcal{K}(\mathcal{N},\mathcal{K})^2}\right). \tag{29}$$

The variables $v_\mathcal{S}(\mathcal{N},\mathcal{S})$, $v_\mathcal{K}(\mathcal{N},\mathcal{K})$, and $v_{\mathcal{FR}}(\mathcal{N},\mathcal{F}_\mathrm{s},\mathcal{R})$ denote the contributions of the sharpness, tonality, roughness, and fluctuation strength, respectively. As can be observed, the loudness contribution is considered in all three terms as it exerts the strongest influence on psychoacoustic annoyance. For the sake of conciseness, the formulation for these three terms is omitted, but the interested reader is referred to (Di et al., 2016).

## 4 Multi-objective optimisation and implementation of the WSE-TSR tracking controller

This section presents an architecture for implementing and optimally calibrating torque control strategies in small-scale VAWTs using the mid-fidelity wind turbine simulation software QBlade (Marten, 2020).

In the following, Section 4.1 formally defines the multi-objective optimisation problem. Section 4.2 explains the multi-criteria decision-making method selected to find the trade-off on the resulting Pareto front. Section 4.3 describes its implementation as a systematic search and guided exploration of the calibration variables for the considered controller, aiming to assess 395 the performance space across all objectives.

### 4.1 Multi-objective optimisation

The present study investigates a multi-objective optimisation problem characterised by a cluster of continuous input variables $\mathcal{X} \subset \mathbb{R}^d$, referred to as the design space (Lukovic et al., 2020). The objective is to minimise an objective function vector, denoted as $\mathbf{f}(\mathbf{x}) = (f_1(\mathbf{x}),\cdots,f_m(\mathbf{x}))$ where $m \geq 2$. In this context, $\mathbf{x} \in \mathcal{X}$ represents the input variable vector, and $\mathbf{f}(\mathcal{X}) \subset \mathbb{R}^m$ denotes 400 the $m$-dimensional image representing the performance space. Thus, the objective is to solve the following minimisation problem, subject to the operating conditions governing the multi-objective optimisation process:

$$\min_{\mathbf{x}}(\mathbf{f}(\mathbf{x})). \tag{30}$$

Since there is an inherent conflict among the objective functions, a single optimal solution may not always exist. Instead, it is necessary to identify optimal solutions, known as the Pareto set $\mathcal{P}_\mathrm{s} \subseteq \mathcal{X}$ in the design space and the Pareto front $\mathcal{P}_\mathrm{f} = \mathbf{f}(\mathcal{P}_\mathrm{s}) \subset$



$\mathbb{R}^m$ in the performance space (Lukovic et al., 2020). In this study, the Pareto front is approximated by considering a point $\mathbf{x}_* \in \mathcal{P}_s$ as Pareto is optimal if there is no other point $\mathbf{x} \in \mathcal{X}$ such that $f_j(\mathbf{x}_*) \geq f_j(\mathbf{x})$ for all $j$ and $f_j(\mathbf{x}_*) > f_j(\mathbf{x})$ for at least one $j$, where $j = \{1, \cdots, m\}$ (Miettinen, 1999).

## 4.2  Multi-criteria decision-making method

From the description of the multi-objective optimisation, it is clear that all points within the Pareto front represent equally
optimal solutions. No solution is better than others in satisfying all conflicting objectives, as enhancing objective function inevitably compromises others (Gambier, 2022; Lara et al., 2023a, b). Once the Pareto front is approximated, the decision-maker can assess various options and select the most favourable one. This collection of potential solutions underscores the adaptability of the design-making process, wherein the designer's role is to identify the optimal solution tailored to specific circumstances (Santín et al., 2017).

To facilitate the decision-making stage, this paper aims to provide designers with a solution to the optimal calibration of the WSE-TSR tracking controller. Therefore, a multi-criteria decision-making (MCDM) method is proposed to select an appropriate trade-off of the considered objective functions. An MCDM method typically involves $p$ alternatives $(A_1, A_2, \cdots, A_p)$ and $q$ criteria $(C_1, C_2, \cdots, C_q)$, structured as a decision matrix $\mathbf{Y} = [y_{c,k}]_{p \times q}$ and weight vector $\mathbf{W} = [w_k]_q$, in which $y_{c,k}$ is the performance of the $c$th alternative with respect to the $k$th criterion and $w_k$ is the weight of the $k$th criterion (Wang et al., 2016).

Simple Additive Weighting (SAW) is applied to each point along the Pareto front, as it is considered the most intuitive and straightforward MCDM approach (Afshari et al., 2010; Bagočius et al., 2014). In the SAW method, the final score of a candidate solution is determined by summing the weighted values of its attributes, accomplished through three sequential steps (Wang et al., 2016). Firstly, the decision matrix $\mathbf{Y}$ is normalised to enable fair comparison across the different criteria, using the Sum method, which is widely applied in the literature (Lee and Chang, 2018). This normalisation yields the normalised decision
matrix $\mathbf{R} = [r_{c,k}]_{p \times q}$. Subsequently, weight values are assigned to each criterion $C_q$ within the weight vector $\mathbf{W}$ (Wang et al., 2016). The next step involves the calculation of the ranking score $S_c$ for each alternative as

$$S_c = \sum_{k=1}^{q} w_k r_{c,k}. \tag{31}$$

The alternative with the highest $S_c$ value is considered the most satisfactory solution (Lara et al., 2023a), and the associated calibration parameters are deemed the most effective trade-off settings for calibrating the WSE-TSR tracking controller.

## 4.3  Implementation of the controller and optimisation framework

In this section, first, the objective functions employed for the multi-objective optimisation of the WSE-TSR tracking controller are defined, followed by a detailed description of the simulation implementation.

### 4.3.1  Definition of the objective function

The approach employed to calibrate the design variables of the WSE-TSR tracking control scheme conforms to the previously
described multi-objective optimisation problem and MCDM method. In this case, a three-dimensional vector captures the





objective functions and is expressed as follows:

$$\mathbf{f}(\mathbf{\Gamma}_d) = [f_1(\mathbf{\Gamma}_d), f_2(\mathbf{\Gamma}_d), f_3(\mathbf{\Gamma}_d)]. \tag{32}$$

The first objective, $f_1(\mathbf{\Gamma}_d)$, relates to the variance of the torque control signal, representing the controller's reactivity and serving as an indicator of the actuation effort on the turbine. This is defined as:

$$f_1(\mathbf{\Gamma}_d) = \frac{\sum_{l=1}^{L}(T_{\mathrm{g},l}(\mathbf{\Gamma}_d) - \bar{T}_{\mathrm{g}}(\mathbf{\Gamma}_d))^2}{L}.$$

The second objective, $f_2(\mathbf{\Gamma}_d)$, encompasses the mean of the wind turbine generated power and is defined as

$$f_2(\mathbf{\Gamma}_d) = -\frac{\sum_{l=1}^{L} P_{\mathrm{g},l}(\mathbf{\Gamma}_d)}{L}.$$

Note that the negative sign preceding the power term is inherent in the context of the minimisation problem defined in the multi-objective optimisation (Equation (30)).

The third objective $f_3(\mathbf{\Gamma}_d)$ concerns psychoacoustic annoyance, quantifying the perceived noise emitted by the wind turbine as

$$f_3(\mathbf{\Gamma}_d) = \mathrm{PA}(\mathbf{\Gamma}_d).$$

These objectives are expected to be conflicting in the sense that a highly responsive controller tends to increase power generation, actuation effort, and noise annoyance. Conversely, a more conservative controller calibration would decrease power
production while being beneficial regarding the actuation and noise objectives.

In the aforementioned equations, the variables are defined as follows: $L$ denotes the entire dataset size, $\bar{T}_{\mathrm{g}}$ represents the generator torque mean value, $T_{\mathrm{g},l}$, and $P_{\mathrm{g},l}$ indicate individual values of generator torque, and power within the recorded data, respectively, and PA is the psychoacoustic annoyance computed using $\alpha$, and BEWS.

It is evident that the resulting signals $T_{\mathrm{g}}$, $P_{\mathrm{g}}$ and PA are dependent on $\mathbf{\Gamma}_d \in \mathcal{X}_d \subset \mathbb{R}^d$, which corresponds to the $d$-dimensional
input variable vector. The current study investigates the input vector dimensionality to evaluate the controller performance under two levels of complexity. The considered input vectors are denoted as

$$\mathbf{\Gamma}_4 = [K_{\mathrm{p,c}}, K_{\mathrm{i,c}}, K_{\mathrm{p,w}}, \lambda_*] \in \mathcal{X}_4,$$
$$\mathbf{\Gamma}_1 = [\lambda_*] \in \mathcal{X}_1,$$

with the subscript $(\cdot)_d$ indicating the dimension of each design space and is used to differentiate between the input vectors
throughout the paper. It should be noted that the WSE-TSR tracking controller is originally formulated by setting $d = 5$ and by denoting the corresponding calibration variables as $\mathbf{\Gamma}_5 = [K_{\mathrm{p,c}}, K_{\mathrm{i,c}}, K_{\mathrm{p,w}}, K_{\mathrm{i,w}}, \lambda_*]$. The selection of a subset from the original formulation is based on the findings reported by (Brandetti et al., 2023b), where it is indicated that the inclusion of an integral term in the estimator (i.e. $K_{\mathrm{i,w}}$) leads to minimal or no enhancement in the operation of the WSE-TSR tracking control scheme. On the other hand, $\mathbf{\Gamma}_1$ represents the $K\omega^2$ controller design space, being one-dimensional as the variation in
$\lambda_*$ results in corresponding changes in the gain $K$, described in Equation (12).



### 4.3.2 Simulation implementation

Figure 8 provides an overview of the overall implementation and optimal controller calibration, which enables the execution of various simulations to explore the parameter space of the considered controllers through a guided search procedure. This process yields a set of optimal solutions that form the Pareto fronts, representing a trade-off between $f_1(\mathbf{\Gamma}_d)$, $f_2(\mathbf{\Gamma}_d)$ and $f_3(\mathbf{\Gamma}_d)$.

The workflow consists of several steps. First, the input parameters, such as the turbine geometry, operating conditions, and torque control strategy, are defined. Then, mid-fidelity simulations are conducted employing the aero-servo-elastic software QBlade (Marten, 2020), loaded as a dynamic link library in MATLAB; the interested reader is referred to the tutorial by Brandetti and van den Berg (2023) for further details about this interface. This interface allows the parallelisation of the original simulation case, referred to as ORIGIN in Figure 8, up to a specified index $t = $ T, significantly reducing the computational time

for the multi-objective controller optimisation. Per simulation, $t$, the controller settings are randomly varied and adhere to the constraints imposed by the design space. In this way, a range of optimal solutions $\mathcal{P}_s^d \subset \mathbb{R}^d$ is explored through a guided search within the constrained design space approximating the Pareto front $\mathcal{P}_f^d = \mathbf{f}(\mathcal{P}_s^d)$.

The VAWT operates in a turbulent wind profile characterised by a mean wind speed of $\bar{V} = 4\,\mathrm{m/s}$ and a turbulence intensity of 15%. The simulation is set over a specific duration; however, for the analysis, only an average of 680 complete turbine

revolutions is considered to eliminate any transient start-up effects from influencing the results. Subsequently, the obtained time series data are used to calculate $f_1(\mathbf{\Gamma}_d)$ and $f_2(\mathbf{\Gamma}_d)$.

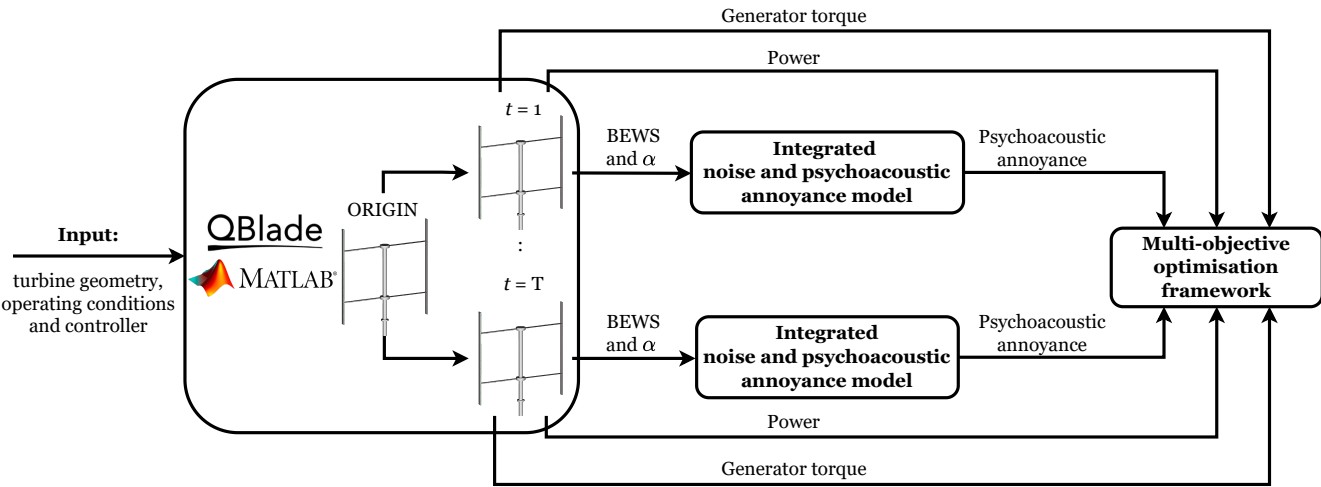

**Figure 8.** Block diagram illustrating the implementation and optimal calibration of controllers applied to the two-bladed $1.5\,\mathrm{m}$ H-Darrieus VAWT. The process involves: defining input parameters, parallelising the ORIGIN simulation case in $t = 1, ..., $ T cases, running simulation $t$ with varied controller gains, extracting aero-servo-elastic information, loading the blade-effective wind speed (BEWS) and the angle of attack ($\alpha$) into the integrated noise and psychoacoustic annoyance model to retrieve the corresponding psychoacoustic annoyance, and using the information within a multi-objective optimisation framework to determine the optimal calibration for the selected controller.





For the computation of the third objective (noise annoyance), $\alpha$ and BEWS are extracted from these time series and employed as inputs for the noise prediction model (Section 3.1). The SPLs for the three noise generation mechanisms described in Section 3.1 are calculated for each time step within one full blade revolution period. Due to the different rotational speeds considered, the rotational period was different for different operational conditions. To consider the total noise emissions of the VAWT, the contributions of the three noise generation mechanisms (LBL-VS, TBL-TE, and T-I) were summed logarithmically for every time step and propagated to the selected observer position. The resulting sound spectra are then auralized, as explained in Section 3.2, to achieve realistic audio files, which are fed into the psychoacoustic annoyance model (Section 3.3), determining $f_3(\mathbf{\Gamma}_d)$. Note that for the considered case study, it was found that $\mathcal{R}$ and $\mathcal{F}_\mathrm{s}$ did not vary significantly. Therefore, a modified version of the PA model, employed in (Merino-Martínez et al., 2022), is applied where these two metrics are not considered, equivalent to setting $\upsilon_{\mathcal{FR}} = 0$. In the following, $f_3(\mathbf{\Gamma}_d)$ is defined as $\mathrm{PA}_\mathrm{mod}$ to denote the modified version.

The last step is to use the computed objective functions $f_1(\mathbf{\Gamma}_d)$, $f_2(\mathbf{\Gamma}_d)$ and $f_3(\mathbf{\Gamma}_d)$ within the multi-objective optimisation framework to determine the optimal calibration for the considered controller.

# 5  Results

This section presents the multi-objective optimisation results. The exploration of the performance space is conducted through a guided search procedure for the group of design variables $\mathbf{\Gamma}_1$ and $\mathbf{\Gamma}_4$, corresponding to the $K\omega^2$ controller and the WSE-TSR tracking controller, respectively. The approximation of the Pareto fronts is based on the minimisation of a weighted linear combination of the objectives $f_1(\mathbf{\Gamma}_d)$, $f_2(\mathbf{\Gamma}_d)$ and $f_3(\mathbf{\Gamma}_d)$, leveraging the data obtained during the exploration process.

The MCDM approach provides a trade-off between the considered objectives, leading to the optimal calibration for both the WSE-TSR tracking controller and the $K\omega^2$ controller. Subsequently, a comparative analysis of the resulting optimal controllers is conducted from two distinct perspectives: the wind turbine performance and the controller performance. Regarding the former perspective, time-domain results are used to evaluate the turbine from an aero-servo-elastic point of view. Additionally, the sound spectra averaged over a rotation are presented in the frequency domain to characterise the acoustic emissions of the VAWT in an urban environment. By extending the analysis beyond conventional performance metrics like power and torque, this study emphasises the critical importance of addressing the impact of noise emissions on psychoacoustic annoyance and its subsequent influence on public perception of VAWTs in urban environments. Lastly, the section uses the frequency-domain framework outlined in Section 2.3 to draw conclusions about controller bandwidth and disturbance rejection performance.

Notably, the analysis focused exclusively on the results obtained from the microphone at $\theta = 90°$, as Brandetti et al. (2023a) demonstrated the VAWT's almost-omnidirectional behaviour in terms of overall sound pressure level when averaging all noise sources.

## 5.1  Pareto fronts and case studies definition

The Pareto front construction begins with systematically exploring the performance space, guided by an investigation of the input variables $\mathbf{\Gamma}$. Figure 9 visually presents the data points obtained from the mid-fidelity simulation scenario. For the higher-



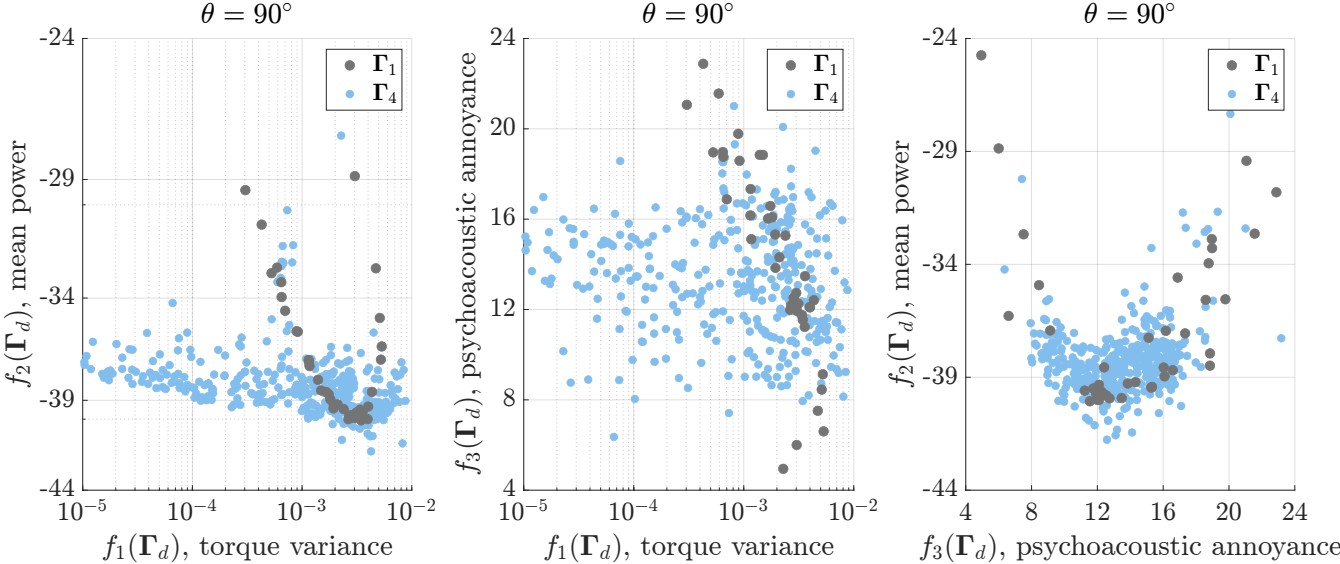

**Figure 9.** Results for the WSE-TSR tracking control scheme performing in turbulent wind conditions attained by an exploratory search of two estimator-controller calibration variables: $\Gamma_1$ and $\Gamma_4$. The controller performance space is defined by the three objective functions $\mathbf{f}(\Gamma_d)$. For the calculation of the psychoacoustic annoyance, the microphone in the x-z plane at $\theta = 90°$ is selected.

dimensional design space $\Gamma_4$, a more extensive dataset is collected to reconstruct the performance space of the WSE-TSR
tracking controller. To facilitate comparative analysis, the $K\omega^2$ controller is used as a benchmark.

Significantly, within the $\Gamma_1$ set, data points demonstrate a clustering pattern with a convex configuration, revealing a distinct global minimum for the objective functions $f_1(\Gamma_d)$ and $f_2(\Gamma_d)$. However, when $f_1(\Gamma_d)$ and $f_3(\Gamma_d)$ are considered as

**Table 2.** Quantitative assessments of the $K\omega^2$ ($\Gamma_1$) and the WSE-TSR tracking control scheme ($\Gamma_4$) for different optimal solutions: ◯, ☆, △, × and □. The % increase is computed for each objective function to show the percentage change of the WSE-TSR tracking controller with respect to the baseline $K\omega^2$. Optimal solutions, such as ◯ and △, demonstrate a substantial reduction in actuation effort alongside increased power production and psychoacoustic changes. However, in some cases like ☆, the controller impact on $f_2(\Gamma_d)$ and $f_3(\Gamma_d)$ remains small. At the same time, trade-off solutions × and □ exhibit interplayed effects on various performance metrics, emphasising the multifaceted nature of controller optimisation.

| Optimal solutions | $f_1(\Gamma_1)$ | $f_1(\Gamma_4)$ | % increase | $f_2(\Gamma_1)$ | $f_2(\Gamma_4)$ | % increase | $f_3(\Gamma_1)$ | $f_3(\Gamma_4)$ | % increase |
|---|---|---|---|---|---|---|---|---|---|
| ◯ | $3.04 \times 10^{-4}$ | $1.04 \times 10^{-5}$ | -97 | -29.42 | -37.05 | 26 | 21.06 | 15.23 | -28 |
| ☆ | $3.4 \times 10^{-3}$ | $4.3 \times 10^{-3}$ | 26 | -40.06 | -41.76 | 4 | 11.54 | 12.58 | 9 |
| △ | $2.3 \times 10^{-3}$ | $6.6 \times 10^{-5}$ | -97 | -24.74 | -34.23 | 38 | 4.95 | 6.36 | 28 |
| × | $3.04 \times 10^{-4}$ | $2.28 \times 10^{-4}$ | -25 | -40.06 | -39.37 | -2 | 11.54 | 12.19 | 6 |
| □ | $3.0 \times 10^{-3}$ | $4.5 \times 10^{-3}$ | 50 | -28.87 | -40.05 | 39 | 6.02 | 9.63 | 60 |



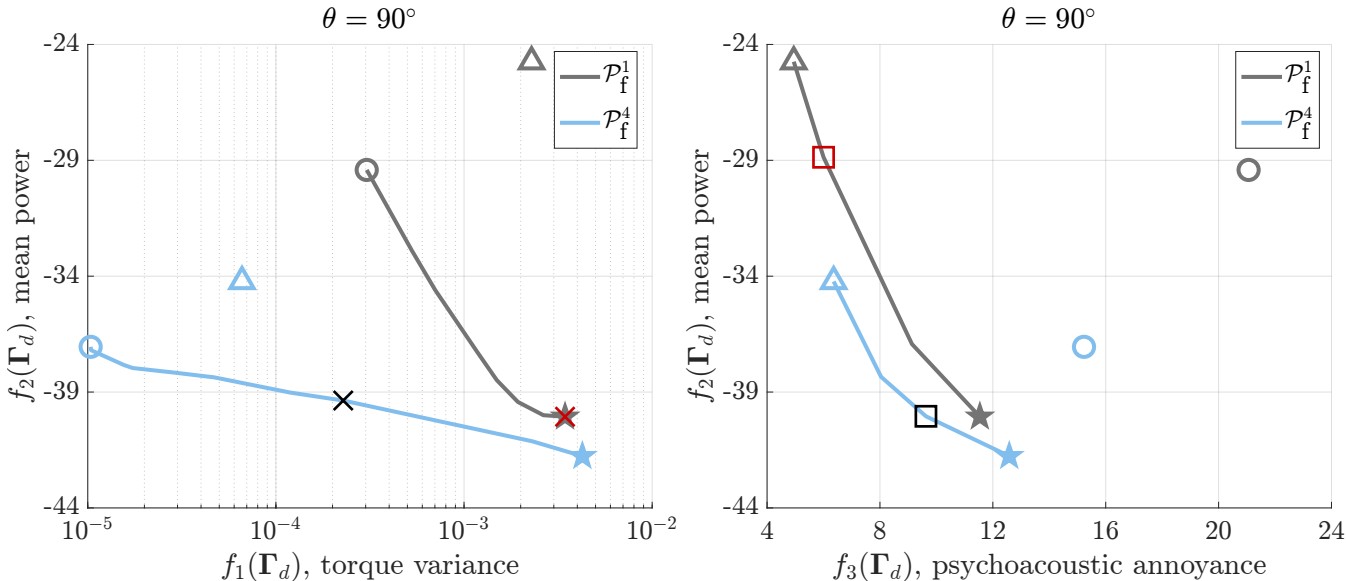

**Figure 10.** Pareto fronts $\mathcal{P}_f^1$ and $\mathcal{P}_f^4$ derived for the $K\omega^2$ and the WSE-TSR tracking controllers under turbulent wind conditions, respectively. For the calculation of the psychoacoustic annoyance, the microphone $\theta = 90°$ is selected in the x-z plane. The optimal solutions for $f_1(\mathbf{\Gamma}_d)$, $f_2(\mathbf{\Gamma}_d)$ and $f_3(\mathbf{\Gamma}_d)$ are indicated using circles (○), stars (☆) and triangles (△), respectively. The trade-off solutions for the two controllers are shown with a cross (×) when $f_1(\mathbf{\Gamma}_d)$ and $f_2(\mathbf{\Gamma}_d)$ are considered as objectives, and with a square (□) when $f_2(\mathbf{\Gamma}_d)$ and $f_3(\mathbf{\Gamma}_d)$ are considered. In contrast to the baseline controller, the WSE-TSR tracking controller achieves improved power maximisation while reducing torque fluctuations and psychoacoustic annoyance, albeit with a slight compromise in power extraction.

objectives, a different pattern emerges. It becomes apparent that the psychoacoustic annoyance does not show a discernible trend with the torque variance, acting as a proxy for the controller actuation effort. Conversely, a clear correlation is observed

between psychoacoustic annoyance and mean power production for both the $\mathbf{\Gamma}_1$ and $\mathbf{\Gamma}_4$ sets, indicating that higher power extraction levels do not necessarily lead to increased psychoacoustic annoyance.

From the available exploration data, the Pareto front is estimated. Figure 10 illustrates the derived Pareto fronts ($\mathcal{P}_f^1$ and $\mathcal{P}_f^4$) for two distinct dimensionalities of the input vector $\mathbf{\Gamma}_d$, facilitating a comparative analysis between the baseline and the WSE-TSR tracking controller performance, respectively. The circles (○), stars (☆) and triangles (△) in the plot represent the

optimal solutions corresponding to each objective function, namely $f_1(\mathbf{\Gamma}_d)$, $f_2(\mathbf{\Gamma}_d)$, and $f_3(\mathbf{\Gamma}_d)$, respectively. Based on the above-mentioned consideration, no Pareto front is constructed between $f_1(\mathbf{\Gamma}_d)$ and $f_3(\mathbf{\Gamma}_d)$ as the exploration points violate the Pareto optimality definition, described in Section 4.1. The trade-offs between $f_1(\mathbf{\Gamma}_d)$ and $f_2(\mathbf{\Gamma}_d)$ and $f_2(\mathbf{\Gamma}_d)$ and $f_3(\mathbf{\Gamma}_d)$ are computed applying the MCDM method defined in Section 4.2 and indicated with crosses (×) and squares (□), respectively.

The figure depicts that the higher-dimensional controller front ($d = 4$) covers the performance space most extensively, con-

firming the effectiveness of the WSE-TSR tracking control scheme in improving the Pareto optimal solutions. Specifically, the controller is capable of minimising psychoacoustic annoyance and torque fluctuations while exerting minimal influence





on power extraction performance. In addition, the application of the WSE-TSR tracking controller to small-scale wind turbines, such as VAWTs in urban environments, leads to attainable power gains. The role of wind turbine inertia in the controller performance is evident, as a higher parameter value enhances resilience against deviations from the optimal operating point,
causing no improvement in power production when applied to large-scale wind turbines (Brandetti et al., 2023b). However, the observed improved power production for the VAWT under study may result in a lower bandwidth than the baseline controller. This aspect will be investigated in the following by applying the frequency-domain framework described in Section 2.3.

The data derived from the Pareto front reveals key insights, as presented in Table 2. This table provides a comprehensive quantitative analysis of the impact of optimal calibration points on system parameters. Specifically, the percentage % increase
is computed for each objective function, showcasing the change in the WSE-TSR tracking controller concerning the baseline $K\omega^2$. When comparing the optimal solutions ◯, the WSE-TSR tracking controller demonstrates a remarkable reduction in actuation effort, up to 97%. This reduction corresponds to a power production increase of 26% and a psychoacoustic annoyance decrease of 28%. As observed, the optimal solution △ also facilitates a reduction in torque fluctuations of up to 97%, accompanied by a 38% increase in mean power and a 28% increase in psychoacoustic annoyance compared to the baseline.
On the other hand, in the case of ☆, the WSE-TSR tracking controller only marginally increases power production by 4%, with a significant increase of 26% in the torque variance and 9% in the psychoacoustic annoyance. A closer examination of the trade-off optimal solution × indicates that the WSE-TSR tracking controller reduces actuation effort by 25%, with a minor impact on power production (only a 2% decrease) and psychoacoustic annoyance (only a 6% increase). Conversely, for the case □, the significant 39% increase in power production is offset by an increase in the psychoacoustic annoyance of 60% and
an increase in torque fluctuations of 50%. The results for the trade-off solutions highlight the complexities of the WSE-TSR tracking controller optimisation.

The following analysis only focuses on the trade-off results × and □ derived from the MCDM approach for the $K\omega^2$ and WSE-TSR tracking controllers. This selective approach aids the decision-making process by providing a clear representation of how these optimal solutions affect wind turbine and controller performance, offering calibration guidelines for the WSE-TSR
tracking control scheme.

## 5.2 Wind turbine results

This section validates the insights obtained from the exploratory search and Pareto fronts by presenting the wind turbine performance results, focusing on aero-servo-elastic performance and acoustic emissions.

### 5.2.1 Aero-servo-elastic performance

The mid-fidelity simulations are conducted in QBlade using the VAWT turbine, as outlined in Section 2.1, under turbulent wind conditions for an urban environment, with a mean wind speed of $\bar{V} = 4\,\mathrm{m/s}$, a turbulence intensity of 15% and a total simulation duration of $800\,\mathrm{s}$. Figure 11 provides a comprehensive representation of the simulation results, encompassing the operating wind speed, tip-speed ratio, generator torque, generator power, tip-speed ratio tracking error, and rotor speed. A condensed representation of the results is included to highlight essential characteristics in the time-domain analysis.





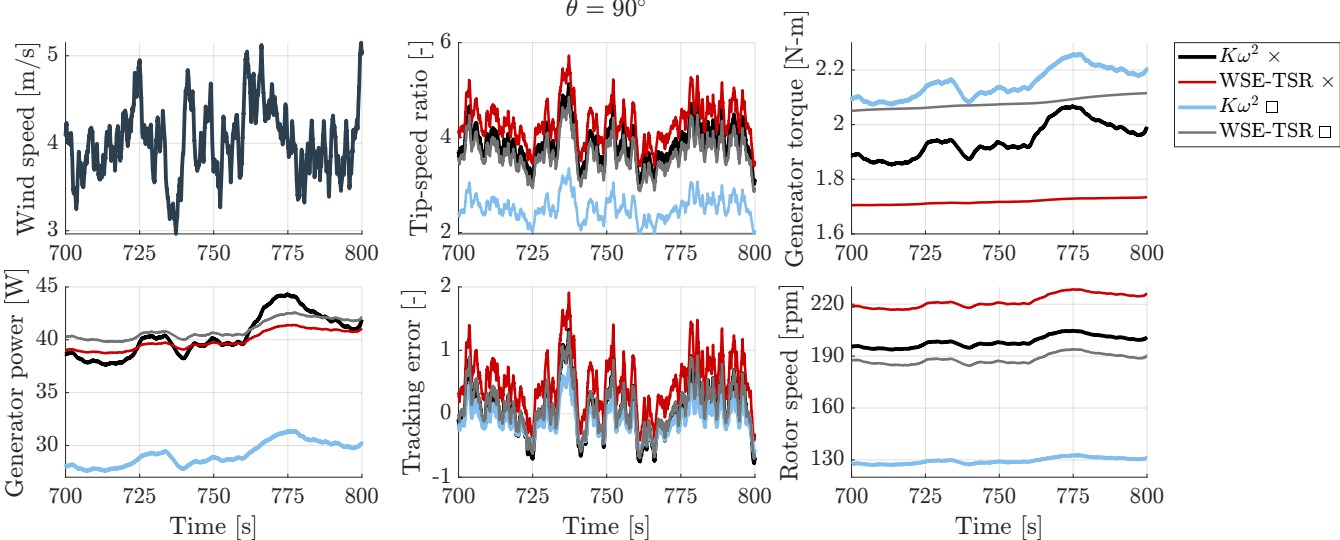

**Figure 11.** Simulation results for the $K\omega^2$ and the WSE-TSR tracking control schemes subject to a turbulent wind speed with a mean of $4\,\mathrm{m/s}$ and a turbulence intensity of 15%. The WSE-TSR tracking controller demonstrates smoother generator curves compared to the fluctuating behaviour of the $K\omega^2$ controller. The $K\omega^2$ trade-off $\square$ case displays a 25% mean power reduction due to suboptimal operation at a reference tip-speed ratio $\lambda_* = 2.6$. Overall, the WSE-TSR tracking controller achieves an optimal balance between reducing psychoacoustic annoyance and maintaining a power output comparable to the maximum power extraction of the baseline control scheme.

For both the selected trade-off case studies of the WSE-TSR tracking controller, the simulations reveal smoother generator torque curves, demonstrating remarkable stability even under turbulent wind conditions. Conversely, the $K\omega^2$ controller exhibits sporadic fluctuations in the generator torque, potentially causing elevated actuation effort and compromising the turbine integrity over prolonged periods of operation. In particular, the trade-off $\square$ case for the $K\omega^2$ controller exhibits a lower mean power than the other three cases, indicating a reduction of over 25%. This difference arises from the controller operating at a

non-optimal reference tip-speed ratio $\lambda_*$ of 2.6 for the power production of the studied VAWT, as illustrated in the power curve of Figure 2. Consequently, the WSE-TSR tracking controller achieves a superior balance between minimising psychoacoustic annoyance and maximising mean production power, enabling comparable power output to the baseline calibrated for maximum power extraction while simultaneously reducing noise levels.

### 5.2.2 Acoustic emissions

The following investigation outlines the different noise generation mechanisms characterising a VAWT in an urban environment by comparing the selected optimal trade-off solutions for the WSE-TSR tracking controller against the baseline control scheme. Given that the psychoacoustic annoyance yields a numerical output, the analysis of noise spectra averaged over a rotation aids in establishing a link between psychoacoustic annoyance and the conventional sound pressure level, thereby effectively characterising the acoustic emissions of a VAWT. In particular, the A-weighted Sound Pressure Level (SPL$_\mathrm{A}$) is employed





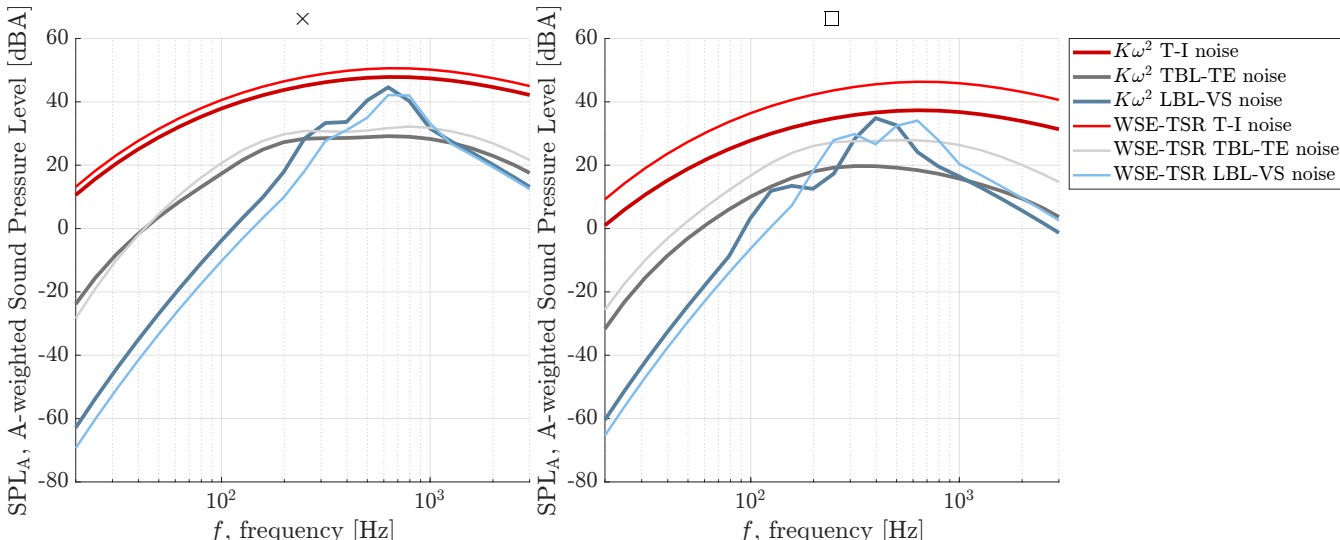

**Figure 12.** A-weighted Sound Pressure Level (SPL$_{\text{A}}$) versus frequency ($f$) obtained from a microphone in the x-z plane at $\theta = 90°$. The $K\omega^2$ controller and the WSE-TSR tracking controller are calibrated for obtaining a trade-off between mean power and torque variance, × (left), and a trade-off between mean power and psychoacoustic annoyance, □ (right). Overall, the most dominant noise source is the T-I noise.

to account for the relative loudness perceived by human hearing, albeit acknowledging that the insights provided are general and highly averaged (Merino-Martínez et al., 2021). The SPL$_{\text{A}}$ is measured in dBA and computed with the noise model of Section 3.1 at a radial distance of $2.6D$ from the centre of the VAWT in the x-z plane, specifically at $\theta = 90°$.

By looking at the different SPL$_{\text{A}}$ spectra in Figure 12 for the considered trade-offs, it is clear that the most dominant noise source is the T-I noise. The SPL$_{\text{A}}$ spectra further support the previous observations derived from the exploratory search and

the construction of the Pareto front. Specifically, for the case ×, the minimal difference in psychoacoustic annoyance is recognised in almost overlapping spectra. Conversely, in the case asterisk, the optimal WSE-TSR tracking controller shows higher psychoacoustic annoyance, reflected in higher SPL$_{\text{A}}$ levels across all three sources. For both cases, the differences between the controllers are most pronounced for the LBL-VS noise source. This is because the controllers lead to different trends in the angle of attack assumed by the wind turbine, which then leads to different alpha-dependent functions (Equation (22)) employed

in the noise model to estimate this source.

### 5.3 Analysis of the controller performance

The current section presents the frequency-domain characteristics of the designated cases, employing the linear analysis framework defined in Section 2.3 (Brandetti et al., 2023b). The analysis presents the frequency responses of the transfer functions $T_{\Lambda_* \to \Lambda}(s)$ and $T_{\mathcal{V} \to \Lambda}(s)$, indicating the performance of the closed-loop system in terms of reference tracking (complementary

sensitivity) and disturbance rejection (sensitivity), respectively. Bode plots are illustrated in Figure 13 for the MCDM solutions.

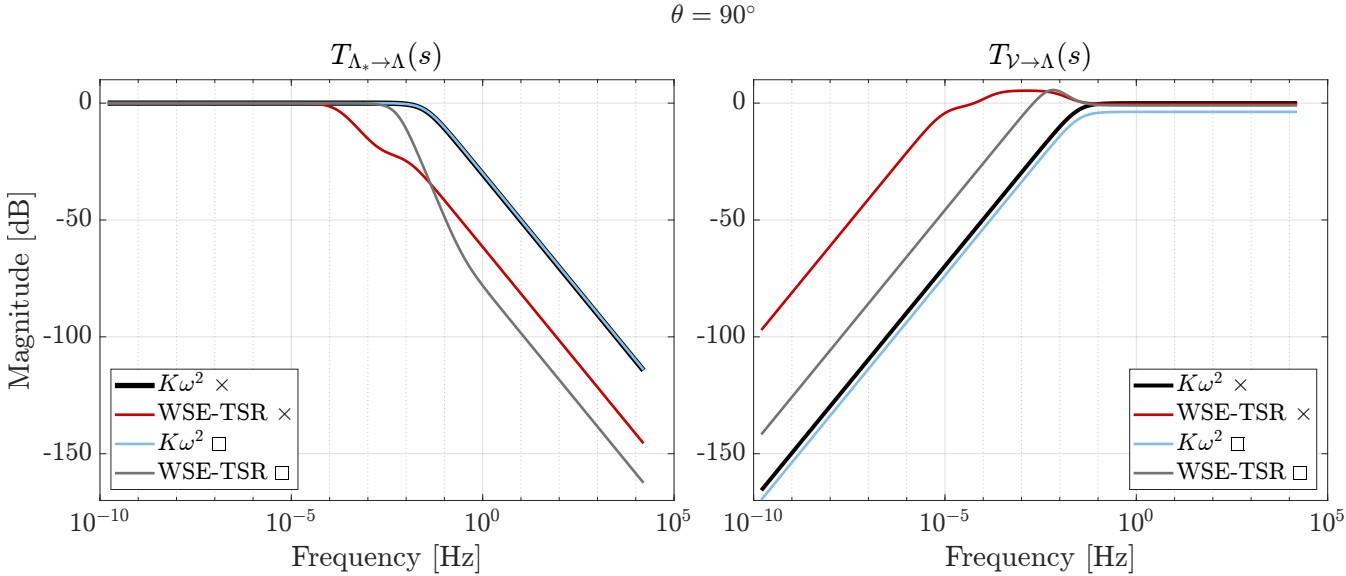

**Figure 13.** Bode plots of the closed-loop transfer functions $T_{\Lambda_* \to \Lambda}(s)$ and $T_{\mathcal{V} \to \Lambda}(s)$ for the baseline $K\omega^2$ and the WSE-TSR tracking controllers. The MCDM solutions for the WSE-TSR tracking controller reveal that the reference tip-speed ratio tracking and disturbance rejection capabilities have to be offset to achieve 39% increase in power production, for the case ×, and 25% reduction in actuation effort, at the cost of 2% power decrease and 6% increase in psychoacoustic annoyance, for the case □, compared to the $K\omega^2$ controller.

In the case of the trade-off $K\omega^2$ ×, the steady-state gain diverges with respect to the baseline gains due to the reference tip-speed ratio $\lambda_*$ calibrated at a lower and non-optimal value of 2.6. On the other hand, the two optimally calibrated WSE-TSR tracking controllers do not exhibit an improvement in control bandwidth but do demonstrate enhanced disturbance-rejection capabilities with respect to the baseline controller. This outcome aligns with the anticipated behaviour, as these
optimal calibrations represent a balance between competing objectives, prioritising an integrated decision-making approach over mere power performance maximisation.

Furthermore, the results underscore the complex trade-offs inherent in the multi-objective calibration of the WSE-TSR tracking controller. As illustrated in Table 2, the improvements in reference tracking and disturbance rejection performance must be offset to achieve advancements in the considered performance metrics. For instance, the case □ demonstrates a remarkable
39% increase in power production, while the × solution allows a significant 25% reduction in torque actuation effort, albeit at the cost of a mere 2% power decrease and a 6% increase in psychoacoustic annoyance.

## 6  Conclusions

This study tackles crucial barriers to the acceptance of small-scale VAWTs in urban environments. Recognising the promising potential of VAWTs for urban wind energy generation, owing to their simple design, low maintenance costs, and reduced
visual impact compared to HAWTs, the study emphasises the need to mitigate noise emissions to overcome local opposition.





Specifically, the research explores the issue of noise annoyance, highlighting the need to incorporate psychoacoustic annoyance in the design and decision-making process to enhance community acceptance of VAWTs. Simultaneously, it underscores the importance of optimising VAWT torque control strategies to maximise the aero-servo-elastic performance of the turbine. By solving a multi-objective optimisation problem, an advanced control strategy is calibrated to achieve the trade-off between the
considered operational performance and noise emissions.

In this study, the combined WSE-TSR tracking controller is employed, renowned for achieving flexible trade-offs in terms of power maximisation and load minimisation. This advanced controller is compared to the baseline $K\omega^2$ control strategy. By employing a multi-objective optimisation approach based on Pareto front approximation and a multi-criteria decision-making method, this paper identifies optimal solutions for the WSE-TSR tracking controller to effectively address the balance between
power extraction, actuation effort, and psychoacoustic annoyance. By analysing these optimal solutions using a frequency-domain framework and mid-fidelity time-domain simulations, the study reveals the significant potential of the optimally calibrated WSE-TSR tracking controller. The controller can decrease the actuation effort up to 25% at the expense of only a 2% decrease in power and a 6% increase in psychoacoustic annoyance in the small-scale urban VAWT under study compared to the baseline. Moreover, the findings underscore the flexible structure of the calibrated controller to balance the aero-servo-elastic
performance with noise emissions effectively.

As regards the noise impact, the T-I noise source is shown to be the dominant noise source for a VAWT in an urban environment. Characterisation of the noise spectra enables a comprehensive understanding of the noise sources contributing to high levels of psychoacoustic annoyance, revealing that increased power extraction levels do not necessarily translate to increased psychoacoustic annoyance.

The findings demonstrate the potential of the proposed methodology, integrating a novel metric for psychoacoustic annoyance into a multi-objective controller optimisation. This comprehensive framework allows multifaceted challenges associated with VAWT deployment in urban environments to be addressed, thereby promoting their acceptance and effective implementation. Future research will be focused on further refining the estimation of psychoacoustic annoyance by performing listening experiments and experimental acoustic measurements on the turbine under study.

*Code and data availability.*   Code and data are available at https://doi.org/10.4121/34b8d260-049a-4f7c-b3cd-60f1f4019696

*Author contributions.*   LB: conceptualisation, methodology, software, validation, investigation, visualisation, writing (original draft). SPM: conceptualisation, methodology, supervision, investigation, writing (review and editing). RM: conceptualisation, methodology, supervision, investigation, writing (review and editing). SW: resources, supervision, writing (review). JWvW: conceptualisation, methodology, supervision, resources, writing (review).



*Competing interests.* At least one of the (co-)authors is a member of the editorial board of *Wind Energy Science.*



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
