# Peer review of "Multi-objective calibration of vertical-axis wind turbine controllers: balancing aero-servo-elastic performance and noise"

_Wind Energy Science, 2023_

## Author Response (AR1)

**Multi-objective calibration of vertical-axis wind turbine controllers: balancing aero-servo-elastic performance and noise**

Livia Brandetti[1,2], Sebastiaan Paul Mulders[2], Roberto Merino-Martinez[3], Simon Watson[1], and Jan-Willem van Wingerden[2]

[1]*Flow Physics and Technology, Faculty of Aerospace Engineering, Delft University of Technology, Delft, The Netherlands*

[2]*Delft Center for Systems and Control, Faculty of Mechanical Engineering, Delft University of Technology, Delft, The Netherlands*

[3]*Aircraft Noise and Climate Effects, Faculty of Aerospace Engineering, Delft University of Technology, Delft, The Netherlands*

Email: l.brandetti@tudelft.nl

The authors appreciate the time and effort the reviewers dedicated to providing feedback on our paper. We are grateful for the insightful comments and valuable improvements to our manuscript. Please see below a point-by-point response to the reviewers' comments in blue, while any changes are highlighted in red. All page numbers refer to the revised paper file.

**Reviewers' comments to the authors:**

**Reviewer 1**

The authors have presented an extensive framework for the optimisation of VAWTs control in the urban environment, accounting not only for the classic performance indicators, but also for novel psycho-acoustic metrics. This is an important step to address the social acceptance issues of such installations. The reviewer believes that the topic and the activity are very interesting, innovative and worthy of investigation. Despite the extensiveness and complexity of the investigation, the authors have managed to maintain an adequate and consistent approach throughout the whole activity. Only the paper structure and presentation need improvement: more in detail, the level of detail given to each element is not balanced. Some specific considerations:

1) Abstract: it is not clear what the elements of novelty of the paper are. Is it the WSE-TSR controller, the optimisation framework or the application to VAWTs?

**Author response:** Thank you for your feedback. Based on your suggestion, we have revised the abstract to provide greater clarity on the elements of novelty in the paper. The revised abstract now explicitly highlights the unique contributions of the research. Specifically, we have highlighted the following key contributions:

1) Formulating and solving a multi-objective optimisation problem dedicated to determining the optimal calibration of the WSE-TSR tracking controller. This calibration is designed as a balanced trade-off between acoustic and aero-servo-elastic performance for an urban VAWT.

2) Integrating a psychoacoustic annoyance metric with an accurate and computationally efficient low-fidelity noise prediction model.

We trust that these changes provide a clearer understanding of the distinctive aspects of our work.

2) Introduction: based on the my experience, the third challenge for small VAWTs is the reliable estimation of their performance

**Author response:** Thank you for observing the third challenge for small VAWTs. We agree that the reliable estimation of their performance poses a significant challenge. We have incorporated this concern into the introduction in response to your suggestion. The revised lines now emphasise the importance of optimising the controller for optimal performance and the reliable estimation of performance in turbulent and fluctuating wind conditions.

Line 40: "However, three main challenges remain for the urban deployment of small-scale VAWTs."

Lines 54-59:" Turning to the second and the third challenges, optimising the controller to ensure an optimal and reliable estimation of the performance of small-scale VAWTs in turbulent and fluctuating wind conditions is paramount (Eriksson et al., 2008; Watson et al., 2019; Bianchini et al., 2022). The combined wind speed estimator and tip-speed ratio (WSE-TSR) tracking controller (Bossanyi, 2000) has been successfully applied to maximise the energy capture of VAWTs (Eriksson et al., 2013; Bonaccorso et al., 2011), demonstrating good dynamic performance in tracking the optimal operating point in turbulent wind conditions and in predicting the turbine performance."

3) Introduction, Line 70-85: this part is quite convolute and hard to understand. Please rewrite for clarity

**Author response:** Thank you for your comment. We appreciate your input and have carefully considered your comments regarding the clarity of lines 70-85 (lines 70-87 of the revised paper).

In response to your feedback, we revised this section to enhance readability and streamline the content. The modifications aim to simplify the language and improve overall comprehension. We believe these changes will address your concerns and make the content more accessible to our readers.

4) Section 2: the description of the test case, including Figs. 1-3, should have a dedicated Section.

**Author response:** Thank you for your suggestion regarding Section 2. We agree with your recommendation. We have now incorporated a dedicated section focused on the wind turbine and its linearised dynamics. This new section comprehensively describes the test case, including Figs. 1-3, to enhance clarity and coherence.

Furthermore, to ensure a well-organised presentation, we have introduced a separate section, i.e. Section 3, that exclusively addresses the wind turbine controllers and the analysis framework.

We believe these adjustments significantly improve the manuscript's overall structure and enhance the reader's experience.

5) The description of the control and noise models is too detailed for the present application. I suggest transforming Sections 2.2, 2.3, and 3.1 into Appendices to improve readability and help the reader focus on the core of the study.

**Author response:** Thank you for your comment. We have carefully considered your suggestion to move Sections 2.2, 2.3, and 3.1 into Appendices for improved readability. While we agree with your recommendation to transfer Section 3.1 to the Appendix (i.e. Appendices A and B), we believe that Sections 2.2 and 2.3 play a crucial role in providing a comprehensive understanding of our methodology and are essential for interpreting the results accurately.

Sections 2.2 and 2.3 delve into the control models, laying the foundation for the subsequent analyses presented in the manuscript. These sections contribute to the clarity and transparency of our study by offering a detailed insight into the methodologies employed. We believe that retaining these sections in the main body of the manuscript is essential for readers to grasp the nuances of our approach and understand the results.

6) Sections 3.2 and 3.3 should be described instead in more detail

**Author response:** Thank you for your comment. We have included additional information in sections 3.2 and 3.3 (now 4.2 and 4.3 with the new numbering) to provide a more detailed explanation of the topics of auralization and psychoacoustic annoyance. To avoid making the manuscript too lengthy for interested readers, we have also added several recommended references for further details. We hope that this satisfies the reviewer's comment.

7) Section 3.1: the noise model receives as an input the boundary layer thickness on the airfoil surface. How is this information extracted from the QBlade model?

**Author response:** Thank you for your comment. The boundary layer parameters needed in the Brooks, Pope and Marcolini (BPM) model are determined analytically using as inputs the blade-effective wind speed and the angle of attack extracted from the QBlade model. The following equations are implemented using the reference (Brooks et al., 1989):

The boundary layer thickness at the pressure side, used in the calculation of the Laminar Boundary Layer-Vortex Shedding (LBL-VS) noise, is

$$\frac{\delta_p}{\delta_0} = 10^{[-0.04175\alpha + 0.00106\alpha^2]}$$

The subscript 0 for the thickness indicates that the airfoil is at zero angle of attack. The corresponding value is computed as:

$$\frac{\delta_0}{c} = 10^{[1.6569 - 0.9045 \log R_c + 0.0596 (\log R_c)^2]}$$

The boundary layer displacement thickness at the pressure side, used in the calculation of the Turbulent Boundary Layer-Trailing Edge (TBL-TE) noise, is:

$$\frac{\delta_p^*}{\delta_0^*} = 10^{[-0.0432\alpha + 0.00113\alpha^2]}$$

The boundary-layer displacement thickness at the suction side, used in the calculation of the Turbulent Boundary Layer-Trailing Edge (TBL-TE) noise, is:

$$\frac{\delta_s^*}{\delta_0^*} = \begin{cases} 10^{0.0679\alpha} & 0° \leq \alpha \leq 7.5° \\ 0.0162(10^{0.3066\alpha}) & 7.5° < \alpha \leq 12.5° \\ 52.42(10^{0.0258\alpha}) & 12.5° < \alpha \leq 25° \end{cases}$$

For both equations, the subscript 0 for the displacement thickness indicates that the airfoil is at zero angle of attack. The corresponding boundary layer displacement thickness results as

$$\frac{\delta_0^*}{c} = 10^{[3.0187 - 1.5397 \log R_c + 0.1059 (\log R_c)^2]}$$

Note that $R_c$ is the Reynolds number based on the chord and is computed using the BEWS and the length of the blade chord $c$. As $R_c$ increases, the thicknesses decrease.

To further clarify this aspect, the following lines have been added in the revised paper, in particular in the Appendix corresponding to the airfoil-self noise model:

Lines 644 to 646: "Note that the boundary layer parameters in the BPM model are computed analytically using as inputs the BEWS and the angle of attack extracted from QBlade. For a detailed description of the equations involved, interested readers are directed to Brooks et al. (1989)."

8) No indication is given in the paper about the QBlade turbine model. What polar data has been used? What aerodynamic sub-models, in particular dynamic stall, have been used? Has the QBlade model accuracy been assessed with respect to higher fidelity load data? Please address this aspects in a dedicated section.

**Author response:** Thank you for your valuable feedback. We acknowledge the importance of providing detailed information about the QBlade turbine model, including the polar data and aerodynamic sub-models, particularly dynamic stall. In response to your query, we have incorporated a dedicated section (Appendix C) in the manuscript to address these aspects. Specifically, we have included comprehensive details about the QBlade turbine model,

including the polar data utilised and the aerodynamic sub-models employed, with a specific focus on dynamic stall.

To further validate the accuracy of the QBlade turbine model, we have included Figure 1, which was made specifically for this review response. This figure illustrates the validation against the Actuator Cylinder model with the Beddoes Leishman dynamic stall model and referenced experimental data (LeBlanc and Simão Ferreira, 2021, 2022). We want to highlight that the experimental forces are derived from a phase-locked average based on approximately 200 data points per azimuthal location. To maintain data integrity, azimuthal locations with poor signal-to-noise ratios have been omitted from the plot.

The averaged loads from the experiment consider the entire rotor span, incorporating 3D tip effects and vortex shedding. Notably, these 3D effects are not accounted for in the Actuator Cylinder model due to its 2D formulation. However, QBlade, as a mid-fidelity simulation solver, does partially consider these effects. The comparison between methodologies reveals a good agreement in terms of trends and absolute values.

In the downwind region (180° < θ < 360°), discrepancies are primarily attributed to the inaccurate prediction of induction and wake effects (3D) in the low and mid-fidelity methods. We emphasize that our primary objective is not to achieve an exact match with experimental data but to present a framework that is accurate enough for design stages. We appreciate your attention to these critical aspects, and we believe that the additional information provided in Appendix C enhances the comprehensiveness and robustness of our study.

[Figure]

*Figure 1 - Phase-locked normal force on blade 1 Fn versus the azimuthal angle ϑ during one turbine rotation. The QBlade results are compared with experimental data and results from the Actuator Cylinder (AC) model.*

The manuscript can be accepted after points 1-8 have been addressed by the authors.

**Reviewer 2**

This manuscript describes an interesting and important study of small wind turbine noise that has significant novelty. The novelty consists of two main elements; the first is stated in the manuscript, but the second is only implied.

The analysis of turbine noise from the perspective of acoustic annoyance is a significant advance for small turbines as no previous studies of which I am aware has attempted anything similar. To consider the noise in the context of control system design and operation provides the second element of the novelty: the study of a small wind turbine as a system. It may surprise readers experienced in large wind turbine research and development, that whole-system dynamics is not a common objective in small wind turbine research.

The manuscript is well organized and well written. The analysis is clearly presented and the multi-dimensional optimization technique may well be a useful contribution on its own. The results look solid and important for small operation in urban areas where noise is critical. I have several small comments that may improve the manuscript. Since I will refer to my own work, any response is not obliged to add the references:

1) The coverage of the noise literature of small wind turbines could be improved. Based on the work of Zhu et al. (2005, additional reference below), similar noise models for the aerodynamic noise to those used here have been developed by Clifton-Smith (2010), Sessarego & Wood (2015, 2022) and Pourrajabian et al. (2023) for small horizontal-axis wind turbines and fans. The last four references include noise models in multi-dimensional optimizations of small turbine design.

**Author response:** Thank you for your valuable feedback. We appreciate your suggestion to improve the coverage of the noise literature in small wind turbines. In response, we have carefully examined the relevant literature and incorporated additional references to address this concern. Specifically, we have included references to the work of Clifton-Smith (2010), Sessarego & Wood (2015, 2022), and Pourrajabian et al. (2023) in the introduction, specifically in Lines 80 to 87.

The revised text now reads as follows:

"Hence, validating the above-mentioned hypothesis on a small-scale wind turbine, like an urban VAWT, holds a significant interest. In the existing body of literature, numerous studies have investigated the multi-faceted aspects of small-scale turbine optimisation, particularly concerning the trade-off between minimising noise emissions and maximising power performance on HAWTs (Clifton-Smith, 2010; Sessarego and Wood, 2015; Pourrajabian et al., 2023). Despite this interest, there is a distinct lack of corresponding studies addressing these aspects in the context of VAWTs. Therefore, this paper tackles the multi-objective optimisation problem from a control perspective by balancing aero-servo-elastic turbine performance (power capture and actuation effort) with noise (psychoacoustic annoyance) for an urban VAWT. Finding a balance between these objectives will further promote the application of VAWTs in urban environments."

We believe that these additions strengthen the coverage of the noise literature and provide a more comprehensive overview of the relevant studies in the field.

2) The aerodynamic model is a quasi-steady one in which power is a function only of tip speed ratio and wind speed. It is well known that, even in steady, uniform flow, vertical axis turbines have a cyclic torque variation which may contribute low frequency noise. In addition, the basic dynamic equation (2) ignores any resistive torque in the generator and drive train, Vaz et al. (2018), which is likely to be higher when a gearbox is used.

3) The noise model considers only aerodynamic sources. In line with the references just cited, any mechanical or electrical component of the noise is ignored.

**Author response:** Thank you for your insightful comments on our work. Your concerns are valid, and we have addressed them in our manuscript.

Regarding the aerodynamic model, we would like to clarify that the gearbox ratio is set to 1 in our simulations. This deliberate choice eliminates resistive torque in the generator and drive train, as highlighted in Vaz et al. (2018). This decision is consistent with our objective to focus solely on the aerodynamic aspects, simplifying the model to specifically examine the dominant aerodynamic noise sources, as these are expected to dominate the far-field noise emissions in most practical situations.

In response to your observation about the cyclic torque variation in vertical axis turbines, we acknowledge this phenomenon even in quasi-steady models. However, for the purpose of our study, the simplified model suffices as it allows us to isolate and analyse the aerodynamic noise mechanisms. We have incorporated this explanation into the revised text.

Our revised text now explicitly states in lines 282 to 284: " The noise model exclusively accounts for aerodynamic sources, excluding any influence from mechanical or electrical components, as aerodynamic noise is deemed dominant for these turbines." Furthermore, in lines 606 to 609, we added, " While the current noise model focuses solely on aerodynamic sources, omitting consideration of mechanical and electrical noise, future iterations of this study hold the potential for extension to encompass these components. Such an expansion would facilitate a comprehensive controller calibration of urban VAWTs in addressing the broader spectrum of noise sources."

We believe these revisions enhance the clarity and transparency of our methodology while acknowledging the limitations.

These are minor comments on a very impressive study.

Additional References

Clifton-Smith, M. J. (2010). Aerodynamic noise reduction for small wind turbine rotors. Wind Engineering, 34(4), 403-420.

Pourrajabian, A., Rahgozar, S., Dehghan, M., & Wood, D. (2023). A comprehensive multi-objective optimization study for the aerodynamic noise mitigation of a small wind turbine. Engineering Analysis with Boundary Elements, 155, 553-564.

Sessarego, M., & Wood, D. (2015). Multi-dimensional optimization of small wind turbine blades. Renewables: Wind, Water, and Solar, 2(1), 1-11.

Sessarego, M., & Wood, D. (2022). Using Small Wind Turbine Technology to Design an AntiFrost Fan. In Journal of Physics: Conference Series (Vol. 2265, No. 4, p. 042071).

J.R. P Vaz, D.H. Wood, D. Bhattacharjee, E.F. Lins (2018) Drivetrain resistance and starting performance of a small wind turbine Renew. Energy, 117, 509-519.

Maillard, J., Bresciani, A.P.C., and Finez, A. (2023). Perceptual validation of wind turbine noise auralization, Proceedings of the 10th Convention of the European Acoustics Association (Forum Acusticum),

Zhu, W., Heilskov, N., Shen, W., and Sørensen, J. (2005) Modeling of aerodynamically generated noise from wind turbines. Journal of Solar Energy Engineering, 127:517-528.